# DeepPhospho accelerates DIA phosphoproteome profiling through in silico library generation

Ronghui Lou 1,2,3,6, Weizhen Liu4,6, Rongjie Li4, Shanshan Li1, Xuming He 4,5✉ & Wenqing Shui 1,2✉

Phosphoproteomics integrating data-independent acquisition (DIA) enables deep phosphoproteome profiling with improved quantification reproducibility and accuracy compared to data-dependent acquisition (DDA)-based phosphoproteomics. DIA data mining heavily relies on a spectral library that in most cases is built on DDA analysis of the same sample. Construction of this project-specific DDA library impairs the analytical throughput, limits the proteome coverage, and increases the sample size for DIA phosphoproteomics. Herein we introduce a deep neural network, DeepPhospho, which conceptually differs from previous deep learning models to achieve accurate predictions of LC-MS/MS data for phosphopeptides. By leveraging in silico libraries generated by DeepPhospho, we establish a DIA workflow for phosphoproteome profiling which involves DIA data acquisition and data mining with DeepPhospho predicted libraries, thus circumventing the need of DDA library construction. Our DeepPhospho-empowered workflow substantially expands the phosphoproteome coverage while maintaining high quantification performance, which leads to the discovery of more signaling pathways and regulated kinases in an EGF signaling study than the DDA library-based approach. DeepPhospho is provided as a web server as well as an offline app to facilitate user access to model training, predictions and library generation.

1 iHuman Institute, ShanghaiTech University, Shanghai 201210, China. 2 School of Life Science and Technology, ShanghaiTech University, Shanghai 201210, China. 3 University of Chinese Academy of Sciences, Beijing 100049, China. 4 School of Information Science and Technology, ShanghaiTech University, Shanghai 201210, China. 5 Shanghai Engineering Research Center of Intelligent Vision and Imaging, Shanghai 201210, China. 6 These authors contributed equally: Ronghui Lou, Weizhen Liu. ✉email: hexm@shanghaitech.edu.cn; shuiwq@shanghaitech.edu.cn

Protein phosphorylation is a widespread post-translational modification (PTM) that regulates essentially all cellular signaling networks[1]. Mass spectrometry (MS)-based phospho-proteomics has become the method of choice for the genome-wide study of protein phosphorylation and dynamic cell signaling[2]. However, conventional phosphoproteomics based on data-dependent acquisition (DDA) often suffers from limited throughput and low reproducibility due to the current MS sequencing speed and semi-stochastic sampling of DDA[3]. With the advent of data-independent acquisition (DIA) to enable proteome profiling of large cohorts of samples with superior quantification accuracy and reproducibility[4,5], DIA-based phosphoproteomics has emerged as a powerful technol-ogy for cell signaling study[6], proteogenomic characterization of clinical cancer tissues[7] and anti-viral drug discovery[8]. Importantly, a benchmark study by Olsen J et al. has demonstrated that DIA phosphoproteomics achieves a larger dynamic range, higher repro-ducibility of identification, and improved sensitivity and accuracy of quantification than DDA phosphoproteomics[3].

However, the current DIA phosphoproteomic workflow faces a significant limitation which is the need of a high-quality spectral library to be constructed prior to data processing. In almost all reported DIA phophoproteomic analysis, a project-specific DDA library was built through DDA analysis of extensively pre-fractionated or repeatedly injected samples[3,6–9]. Although project-specific DDA libraries afford a higher proteome coverage (i.e., covering a larger number of protein and peptide identifica-tions) than other experimental libraries, they are built at the expense of time, sample, and considerable efforts with pre-fractionation[10]. Alternatively, the previous elegant study showed it is feasible to build a direct DIA library from DIA data alone for deep phosphoproteome profiling. But in that case, the DIA library was constructed based on data acquisition from an unusually large sample cohort (186 DIA runs)[3].

Apart from the experimental DDA and DIA libraries, in silico libraries can be generated through predictions of fragment ion intensity and retention time for given peptide sequences using deep learning or traditional machine learning-based methods[11–14]. Recently developed deep neural networks have been implemented to generate in silico libraries for DIA data analysis in global proteomics, which resulted in a whole-proteome coverage nearly equivalent to or even higher than experimental DDA libraries[10–12,14]. We also reported the construction of a protein family-targeted in silico library with deep learning tools so as to deepen the sub-proteome coverage of a selected protein family[15]. In spite of these promising applications, the in silico library approach has never been explored in DIA phosphoproteomic data mining. Furthermore, the existing deep learning methods mostly adopting a single LSTM or RNN archi-tecture employ a linear embedding of amino acids, followed by a convolutional or recurrent neural network that extracts peptide fea-tures for the prediction. Such a strategy, however, is often limited by the restrictive structural assumption on the designed deep networks, which may cause difficulty in handling peptides of variable lengths or capturing their rich structural properties such as PTMs.

In this study, we first developed a deep learning framework, termed DeepPhospho, to achieve highly accurate predictions for phosphopeptides. Through designing and evaluating a series of in silico libraries generated by DeepPhospho, we demonstrated that DeepPhospho predicted libraries outperform the benchmark experimental DDA library and accomplish faster and deeper DIA phosphoproteome profiling.

## Results

### Principle of DeepPhospho.
The key ingredient of DeepPhospho is the learning of a gradually richer peptide representation, which allows for better capturing the local and global structure of a peptide for fine-grained prediction. In contrast to prior methods, we adopted a hybrid network design that integrated two types of network architectures to encode different aspects of the peptide structure.

To this end, we developed a modular deep network consisting of three main sub-networks: a recurrent network for encoding peptides, a Transformer network for refining the peptide representation, and a regressor network for predicting fragment ion intensities or indexed retention time (iRT) (Fig. 1a, Supplementary Fig. 1a). The main modules of our network were organized in a sequential manner and gradually enhanced the peptide features. Specifically, given the input peptide sequence and optionally the charge state, we first employed a bi-LSTM network to compute an initial representation of all the amino acids in the sequence. The bi-LSTM network embedded each amino acid into a vector representation, which was then updated by two layers of bidirectional LSTM units. This produced a context-aware representation as each amino acid was enriched by the features of other amino acids in the same peptide. However, the effective context encoded by the bi-LSTM network is often limited due to loss of information in its recurrent updates[16]. To capture long-range dependency in the peptide sequences, we then introduced the second module, a Transformer network that refines the peptide representation generated from the first module. The transformer network used a multi-head self-attention to update the features of all amino acids in parallel, and enabled the model to directly attend to multiple sites of the peptide even if they were far apart. Finally, the network output a new representation for the input peptide, which was then fed into a linear regressor network to generate predictions for RT or ion intensities.

To tailor DeepPhospho specifically to phosphopeptide predic-tion, we introduced a set of extra tokens to represent different phosphorylated amino acids, and learn their embedding jointly with the base peptides. In addition, for the task of fragment ion intensity prediction, we designed a modified loss training that enforced the structural constraints of the corresponding peptides. In particular, we ignored the loss terms on the phosphate moiety that cannot exist and filter out the model predictions on those ions.

To the best of our knowledge, DeepPhospho is the first work to utilize the Transformer for the prediction of peptide fragmenta-tion patterns though it has been extensively used in the natural language processing[17,18]. To demonstrate the advantage of our model design, we conducted an ablative study to compare our model with the bi-LSTM or the Transformer alone, and combination of CNN with the Transformer using two phospho-proteomic datasets (Supplementary Data 1). Our hybrid model consistently outperformed those alternative baselines, indicating that DeepPhospho is able to learn a better feature representation for phosphopeptides, and the bi-LSTM and the Transformer are complementary in learning the peptide representation (Supple-mentary Fig. 1b, c).

### Accurate prediction of fragment ion intensity and retention time for phosphopeptides.
After the model architecture test, DeepPhospho was pre-trained using four large-scale phospho-proteomic datasets (details in "Methods"; Supplementary Data 1). We then used DeepPhospho to make predictions for phospho-peptides in three other datasets acquired on Q Exactive HF-X and Orbitrap Fusion Lumos mass spectrometers from two laboratories (Supplementary Data 1). Two datasets (RPE1 DDA and RPE1 DIA) both collected from RPE1 cells[3], one by DDA, the other by DIA acquisition methods, were searched by MaxQuant and Spectronaut respectively to yield phosphopeptide identification

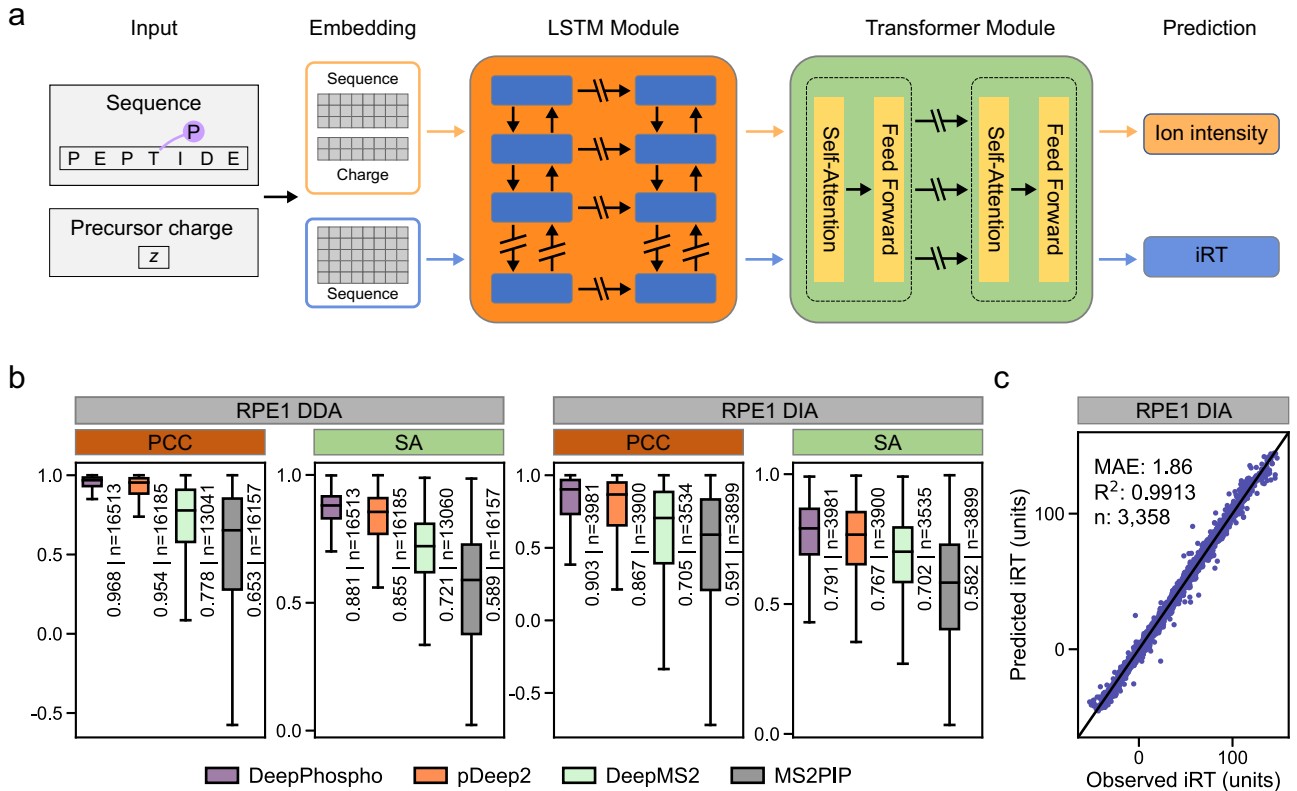

**Fig. 1 Model architecture and performance of DeepPhospho. a** The DeepPhospho deep learning architecture for indexed retention time (iRT) and fragment ion intensity prediction for any given phosphopeptide. Given a peptide sequence and the precursor charge as input, our model first uses a bi-LSTM network to compute an initial representation of all the amino acids, which are then refined by a Transformer module. The resulting global features are fed into a linear regressor network to generate predictions for fragment ion intensity and iRT. **b** Evaluation of DeepPhospho and three other models based on the distribution of Pearson correlation coefficient (PCC) and spectral contrast angle (SA) calculated between predicted and experimental MSMS spectra from two datasets. Median PCC and SA are indicated; n is the number of phosphopeptides in the test set. Boxplot center line, median; box limits, upper and lower quartiles; whiskers, 1.5× interquartile range. **c** Evaluation of DeepPhospho based on the correlation of predicted and experimental iRT values. Correlation coefficient of linear regression ($R^2$) and median absolute error (MAE) are indicated. Source data for this figure are provided as a Source data file.

results. These identification output files were then imported into Spectronaut to generate a DDA library and a direct DIA library which both contained MSMS spectra, PTM site localization, and iRT data for identified phosphopeptides in the same format. Data in each library was split at a ratio of 8:1:1 for model training, validation and test separately. The trained DeepPhospho model achieved excellent overall agreement between the experimental and predicted fragment ion intensities for the test set (median Pearson correlation coefficient (PCC) = 0.968, median spectral angle (SA) = 0.881 for RPE1 DDA; median PCC = 0.903, median SA = 0.791 for RPE1 DIA) (Fig. 1b). Furthermore, DeepPhospho enabled accurate iRT prediction for both datasets after model training (median absolute error (MAE) = 1.74 units for RPE1 DDA, 1.86 units for RRE1 DIA) (Fig. 1c, Supplementary Fig. 2a). For the third dataset (U2OS DIA) which is another DIA library generated from phosphoproteome profiling of U2OS cells[9], DeepPhospho made equally accurate predictions of fragment ion intensity and iRT (Supplementary Fig. 2a, b). Evaluation of model predictions indicated better performance for mono-phosphosite peptides and for phosphopeptides merely containing pS than the other categories possibly because of their larger data fractions available in model training (Supplementary Fig. 2c–e).

We also compared the performance of DeepPhospho in phosphopeptide fragment ion intensity prediction with three recently reported models. pDeep2 built on an LSTM model also allows for transfer learning with a training set[19]. DeepMS2

initially predicts for non-phosphopeptides and generates in silico MSMS spectra for the modified peptides using a "budding" strategy[20]. MS2PIP predicts fragmentation patterns directly from phosphopeptide sequences using an XGBoost machine learning algorithm[21]. In all cases, DeepPhospho outperformed the reported models when tested with the same phosphoproteomic datasets (Fig. 1b, Supplementary Fig. 2b).

Because our major attempt was to enhance DIA data mining, we looked into a sub-population of phosphopeptides with a relatively low correlation (PCC < 0.3) between their library spectra from U2OS or RPE1 DIA data and the predicted spectra (Fig. 2a upper). To determine which spectrum is more likely to be correct for a given phosphopeptide, we obtained the reference spectrum from the profiling results of the same or very similar phosphoproteome samples analyzed by gold-standard DDA acquisition methods[1,9] (Supplementary Data 1). Notably, the majority of phosphopeptides (78.6% in RPE1 DIA data, 82.8% in U2OS DIA data) showed a stronger correlation between the reference and predicted spectra than between the library and predicted spectra (Fig. 2a lower). This result suggests that the fragmentation patterns for these phosphopeptides may be more accurately predicted by DeepPhospho than experimentally assigned in the library.

To verify this finding, we synthesized seven phosphopeptides and acquired the bona fide high-quality MSMS spectra by targeted MS analysis of the synthetic peptides. All predicted

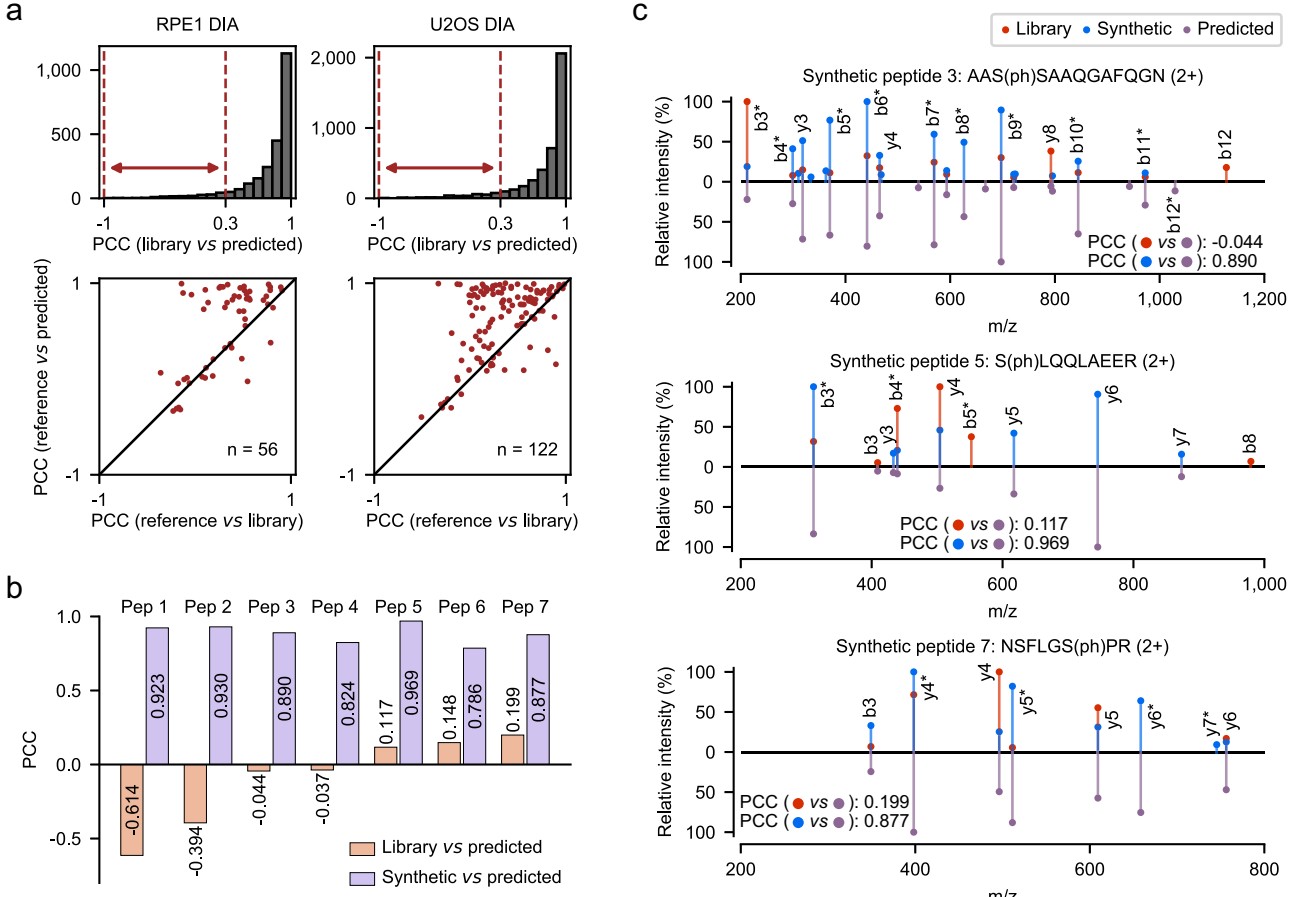

**Fig. 2 MSMS spectra prediction by DeepPhospho pinpoints possibly false identifications in an experimental library. a** Distribution of PCC between spectra predicted by DeepPhospho vs spectra assigned in the DIA library for two datasets (upper). For phosphopeptides of low spectral similarity (PCC within −1 to 0.3), their PCC distribution are calculated between predicted spectra vs reference spectra, and between predicted spectra vs DIA library spectra, and plotted around the diagonal (lower). Reference spectra were obtained by gold-standard DDA analysis of the same phosphopeptide samples. **b** Correlation between the predicted spectra and the high-quality spectra of the synthetic peptide, and between the predicted spectra and the DIA library spectra, for seven selected phosphopeptides. **c** Spectra mirror plots for phosphopeptides show much higher similarity between the predicted spectra, and the synthetic peptide spectra than between the predicted spectra and the DIA library spectra. Relative fragment ion intensities in the predicted spectra, the DIA library spectra and the synthetic peptide spectra are annotated by purple, orange and blue lines. * Indicates the loss of a phosphate. Source data for this figure are provided as a Source data file.

spectra were closely correlated with the bona fide spectra (PCC 0.79–0.97) whereas the DIA library spectra for the same phosphopeptide sequences showed much lower correlations (PCC −0.61–0.20) (Fig. 2b). Mirror plots for specific phospho-peptides also reflected strong agreement between our prediction and bona fide measurement yet considerable discordance with the DIA library spectra (Fig. 2c, Supplementary Fig. 3). Taken together, DeepPhospho enables accurate prediction of fragment ion intensity for phosphopeptides, which in some cases could pinpoint possibly false identifications in an experimental library.

**Constructing DeepPhospho predicted libraries for DIA phosphoproteomics data mining.** In routine DIA data analysis, a project-specific DDA library has to be built based on peptide identifications from a separate DDA experiment[22]. Raw DIA data are then processed for peptide identification and quantification using a peptide-centric scoring algorithm[23] against this experimental DDA library. Particularly, in a DIA phosphoproteomic experiment, to construct a conventional DDA library, one would need to go through an extensive procedure of phosphopeptide preparation, enrichment, pre-fractionation, and LC-MS/MS analysis which takes weeks to months to complete the data

acquisition (Fig. 3a). Alternatively, a direct DIA library can be generated by searching the raw DIA data directly and exploited for DIA data mining. Construction of this DIA library typically requires single-injection DIA data acquisition which can be finished within days, thus largely saving instrument time and precious samples (Fig. 3a). However, up till now, the proteome coverage of a DIA library still lags behind that of an extensive DDA library, which greatly limits the depth of proteome profiling if using the DIA data alone[3,5].

To investigate whether and to what extent in silico spectral libraries can deepen DIA phosphoproteome profiling, we designed six types of predicted libraries or hybrid libraries to be assessed in parallel with the project-specific DDA library namely Lib 1 (Fig. 3b): Lib 2, a predicted DDA library; Lib 3, a hybrid of the direct DIA library and the predicted DDA library; Lib 4, a hybrid of the direct DIA library and the predicted library from a public phosphoproteome database; Lib 5, a hybrid of the direct DIA library and the predicted library from a public phosphosite database; Lib 6, a hybrid of the predicted DIA library and the predicted DDA library; Lib 7, a hybrid of the predicted DIA library and the predicted library from a public phosphoproteome database. Of note, all predicted libraries in Lib 2 to Lib 7 were

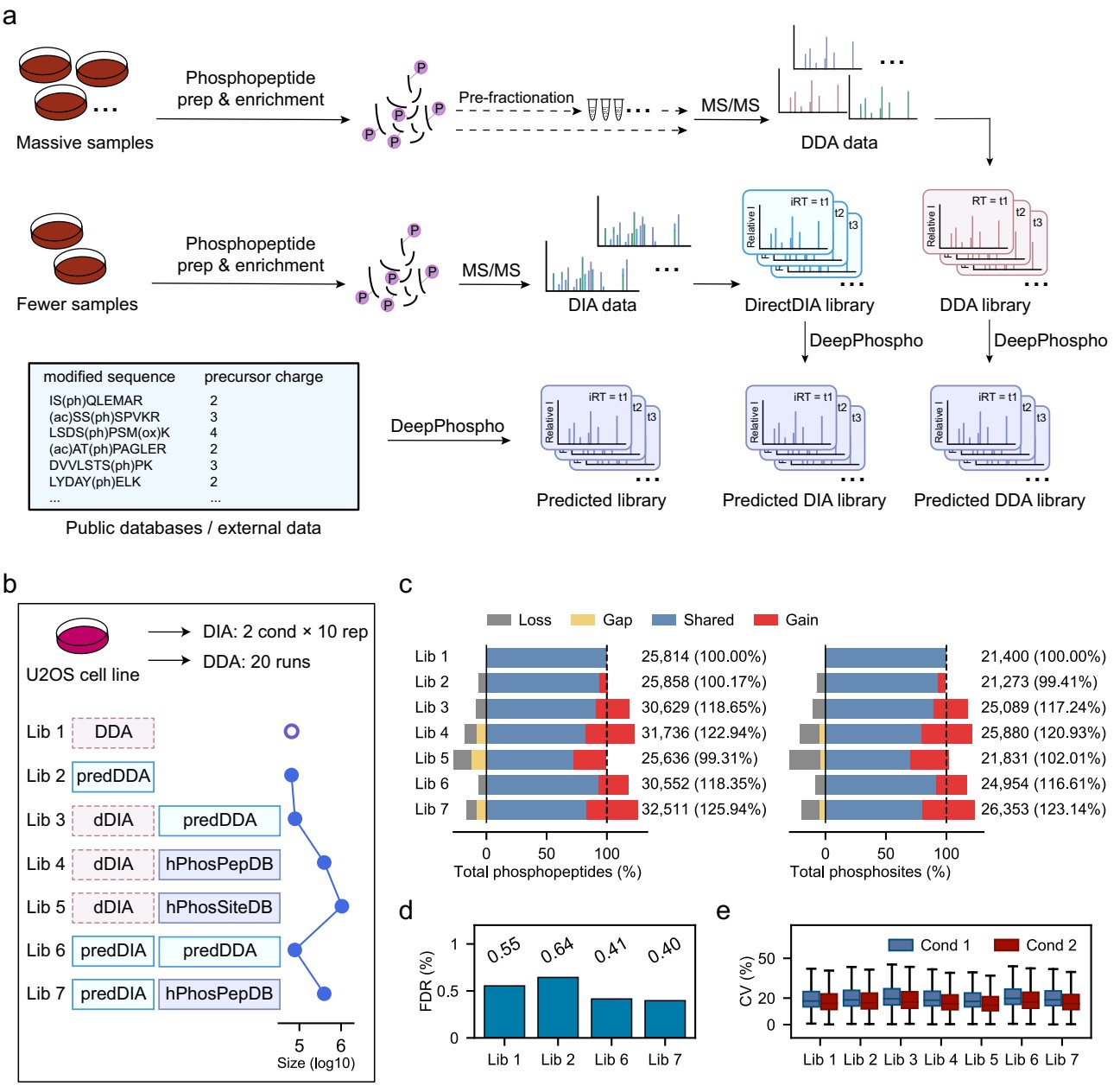

**Fig. 3 Generation of DeepPhospho predicted libraries for DIA phosphoproteomics data mining. a** An experimental DDA library or direct DIA library can be converted to a predicted DDA library or a predicted DIA library by DeepPhospho. A predicted library can be also generated from public phosphoproteome or phosphosite databases, or external phosphoproteomics data. **b** Design of seven spectral libraries for the U2OS DIA data analysis. DDA/dDIA, experimental DDA or direct DIA library; predDDA/predDIA, predicted libraries converted from the DDA and DIA library; hPhosPepDB/ hPhosSiteDB, predicted libraries built from public human phosphoproteome and phosphosite databases. Lib 3 to Lib 7 are comprised of two separate libraries. **c** Number of phosphopeptides and phosphosites identified using each library. Percentage of the total phosphopeptide or phosphosite number is shown for each predicted library relative to the project-specific DDA library (Lib 1). The proportions of shared identifications (IDs), gained IDs, lost IDs, and gap IDs yielded by Lib 2 to Lib 7 compared to Lib 1 are indicated in different color. Gap IDs are those present in Lib 1 yet absent in the DeepPhospho predicted libraries, thus they cannot be identified with the latter. **d** Library-specific FDR assessed using the target-decoy strategy. **e** % coefficient of variation (CV) of all phosphopeptide quantification between 10 replicates at each condition. Boxplot center line, median; box limits, upper and lower quartiles; whiskers, 1.5× interquartile range. Source data for this figure are provided as a Source data file.

generated by DeepPhospho based on the phosphopeptide sequences and charge states recorded in the corresponding experimental libraries or databases (Fig. 3a).

**Comparison of DeepPhospho predicted libraries with the project-specific DDA library.** To evaluate the performance of different DeepPhospho yielded libraries, we first implemented

individual libraries in DIA data mining of the U2OS DIA dataset collected under two conditions (control vs drug-treated, Supplementary Data 1)[9]. In this experiment, a project-specific DDA library was built from 20 DDA runs of the same phosphoproteome samples. After training DeepPhospho with U2OS DIA data, Lib 2 to Lib 7 were generated based on predictions for phosphopeptides identified in the DDA or direct DIA library, and phosphopeptide sequences registered in a human phosphoproteome database[24]

(hPhosPepDB) or computed from a human phosphosite database[25] (hPhosSiteDB) (Fig. 3b and Supplementary Data 2). Specifically, for hPhosPepDB which records 204,606 human phosphopeptides identified in various proteomics projects, we generated 21 predicted libraries depending on the combination of precursor and fragment mass ranges, peptide length, max phosphosite number, and charge state in different values (details in "Methods"). Then the best combination giving rise to the highest phosphoproteome coverage was used to generate Lib 4 and Lib 7 (Supplementary Fig. 4). Meanwhile, 350,719 phosphopeptide sequences were computed through collecting human phosphosites registered in EPSD database and in silico digestion of the human proteome, which were used to generate the predicted library in Lib 5. Notably, Lib 4, Lib 5 and Lib 7 comprised of a predicted library from public databases are of a larger size than the others by one order of magnitude and contain a unique fraction of phosphopeptides not present in Lib 1 (Fig. 3b, Supplementary Fig. 5a, b). All three predicted libraries consist of peptide precursors with charges states of 2/3/4 whereas their sequences and phosphosites are defined by the databases.

Analysis of the U2OS DIA data with each library by Spectronaut led to varying phosphoproteome coverages (Fig. 3c). The largest increase of coverage was attained with Lib 7 which yielded 32,511 phosphopeptide and 26,353 phosphosite identifications, compared to 25,814 phosphopeptides and 21,400 phosphosites originally identified with Lib 1. All phosphosites reported in our study required at least 0.75 localization confidence (Class I sites) as in the previous analyses[3,26]. Lib 7 was produced by merging a small predicted DIA library with a large one predicted from hPhosPepDB. It is noteworthy that Lib 7 gained even more identifications than Lib 6, a hybrid of the predicted DIA and the predicted DDA libraries which also outperformed Lib 1 by covering 30,552 phosphopeptides and 24,954 phosphosites. Interestingly, data analysis with Lib 5 which is mainly comprised of a predicted library from hPhosSiteDB led to no increase of coverage, which underlies the importance of selecting an appropriate database and optimizing the library construction parameters in the performance of predicted libraries built on public databases.

DIA data analysis with two best-performing predicted libraries Lib 7 and Lib 6 led to identifications of 10,987 and 6453 new phosphopeptides as well as localizations of 9177 and 5390 new phosphosites respectively that were absent in the analysis with Lib 1 (Supplementary Fig. 5c). The huge gains prompted us to assess the control of false discovery rate (FDR) even though <1% FDR was automatically set at both peptide and protein levels by Spectronaut in all our data searches. For Lib 1 and three predicted libraries Lib 2, Lib 6, and Lib 7, we created a reverse library by predicting MSMS fragmentation pattern and iRT for the reverse sequence of each identified phosphopeptide in the original library using DeepPhospho. Searching the same dataset with the original library appended with the corresponding reverse library allowed us to assess library-specific FDRs. FDR turned out to be equivalent between Lib 1 (0.55%) and all three predicted libraries (0.40–0.64%) (Fig. 3d, Supplementary Fig. 5d). In addition, we created two-species libraries by merging each predicted library (positive set) with a predicted *Arabidopsis thaliana* phosphoproteome library (negative set), and FDRs were estimated to be 0.85 and 2.05% for DIA data analysis using Lib 6 and Lib 7 (Supplementary Fig. 5e). Thus, significant increase of phosphoproteome coverages by using DeepPhospho predicted libraries did not compromise the FDR control. Furthermore, reproducibility of phosphoproteome quantification between replicates was comparable among Lib 1 and all six DeepPhospho predicted libraries (Fig. 3e).

**Performance of DeepPhospho predicted libraries in a phosphosignaling study**. Next we used the RPE1 DIA dataset from a cell signaling study[3] to evaluate whether the advantage of DeepPhospho predicted libraries in deepening phosphoproteome profiling can be translated to a more biological scenario. In this study, RPE1 cells were stimulated with EGF in the absence or presence of two MEK kinase inhibitors. DIA data from 18 runs were acquired from the phosphoproteome samples prepared under six conditions in biological triplicates (Fig. 4a, Supplementary Data 1). In addition, this study recorded a project-specific DDA library consisting of 89,416 unique phosphopeptides identified from 147 DDA runs to analyze extensively prefractionated samples[3]. After training DeepPhospho model with RPE1 DIA data, we created five predicted or hybrid libraries following the design of Lib 2, Lib 3, Lib 4, Lib 6, and Lib 7 as described above. Lib 5 was abandoned here because of its poor performance in the previous evaluation. For the construction of Lib 4, we also compared 21 different combinations of phosphopeptide and precursor features so as to yield a predicted library from hPhosPepDB with the highest phosphoproteome coverage (Supplementary Fig. 6). Both Lib 4 and Lib 7 comprised of the largest predicted library from hPhosPepDB exceeded Lib 1 in size, with each having a unique fraction of phosphopeptide identifications (Supplementary Fig. 7a). Then the RPE1 DIA data were processed with different libraries to give rise to profiling results.

Given that the biological goal of this study was to map regulated phosphosites at different conditions so as to characterize EGF-dependent phosphosignaling in the context of MEK inhibition[3], we focused on quantifiable phosphopeptides and phosphosites with ratios measured between any two experimental conditions. Not surprisingly, all five DeepPhospho predicted libraries outperformed the extensive project-specific DDA library (Lib 1) by increasing the total number of quantifiable phosphopeptides and phosphosites (Fig. 4b). The winner was Lib 6 which gained 17.9 and 14.9% more quantifications of phosphopeptides and phosphosites relative to Lib 1 (Fig. 4b).

To further deepen the phosphoproteome coverage especially for the quantifiable portion, we explored an iterative search strategy. Phosphopeptides identified from the initial search with a specific library were selected to build a focused library which was used to iteratively search the raw DIA data with the same parameters as the initial search (Fig. 4c). These focused libraries are 3–14-fold smaller than the initial libraries (Supplementary Fig. 7b). Remarkably, the iterative search with all focused libraries significantly increased the coverages of quantifiable phosphopeptides whereas the coverages of totally identified phosphopeptides and non-phosphopeptides remained barely changed (Fig. 4d, Supplementary Fig. 7c). It suggests the iterative search specifically expands the fraction of the quantifiable phosphoproteome within the entire profiled proteome. Relative to Lib 1 which quantified 14,274 phosphopeptides and 12,726 phosphosites, Lib 7 showed the best performance by quantification of 17,366 phosphopeptides (21.7% increase) and 14,994 phosphosites (17.8% increase). We speculate that the iterative search enhances the sensitivity of detecting phosphopeptides identified in the initial search, which results in fewer missing values to facilitate ratio measurement of more peptides. Notably, the same peptides identified in both initial and iterative searches were assigned to the same features, suggested by the perfect correlation of RT measurement between two searches for all co-identified peptides (Supplementary Fig. 8).

We also assessed the FDR control of both initial and iterative searches on this dataset using the original-reverse combined library. Lib 6 and Lib 7 had even smaller error rates (0.29 and 0.41% for initial searches, 0.29 and 0.21% for iterative searches) than Lib 1 (0.71% for initial search, 0.88% for iterative search) (Supplementary Fig. 7d). With two-species libraries, FDRs were

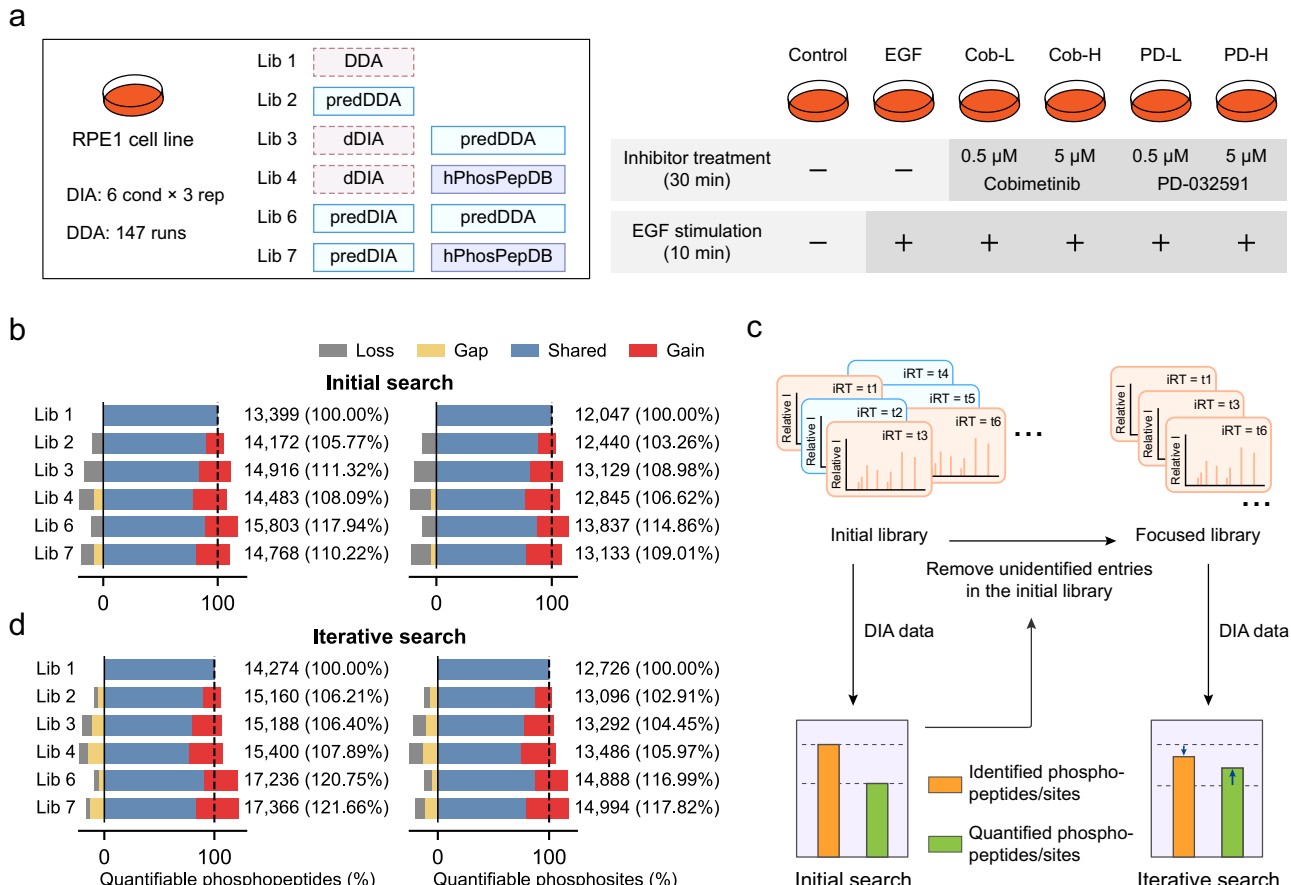

**Fig. 4 DIA data analysis with DeepPhospho predicted libraries in a phosphosignaling study. a** Design of six spectral libraries as defined in Fig. 3b for the RPE1 DIA data analysis (left) and experimental design of the EGF signaling study in the context of MEK inhibition (right). **b** Number of phosphopeptides and phosphosites that were quantified from the initial search using each library. **c** Procedure of building a focused library to be used for an iterative search. **d** Number of phosphopeptides and phosphosites that were quantified from the iterative search using each library. Percentage of the total quantifiable phosphopeptide or phosphosite number is shown for each predicted library relative to Lib 1. The proportions of shared identifications (IDs), gained IDs, lost IDs, and gap IDs yielded by Lib 2 to Lib 7 compared to Lib 1 are indicated in different color. Source data for this figure are provided as a Source data file.

estimated to be 0.06-0.89% for initial and iterative searches using Lib 6 or Lib 7 (Supplementary Fig. 7e).

To precisely assess the false localization rate (FLR) of phosphosites, we further analyzed a DIA data set acquired on 166 synthetic human phosphopeptides containing 176 clearly defined phosphosites[27]. Hybrid libraries were constructed by merging the synthetic phosphopeptide library with a much larger experimental or predicted library. DIA data analysis with the hybrid experimental DDA library gave rise to FLR < 5% in both initial and iterative searches (Supplementary Fig. 9). While data analysis with the hybrid predicted libraries based on hPhosPepDB or hPhosSiteDB yielded FLRs of 6–7% in initial searches, iterative searches with the same libraries reduced the FLR to 2.9 and 3.5%. In addition, iterative searches with two hybrid predicted libraries reached the maximal rate of true phosphosite recovery (94–95%) that could be achieved with the pure synthetic peptide library (Supplementary Fig. 9). Analysis of another DIA data set on 300 synthetic yeast phosphopeptides[28] yielded a similar FLR of 3.0% for the iterative search with the predicted yPhosSiteDB library (Supplementary Fig. 9). Therefore, the synthetic phosphopeptide data analysis implied that the FLR control for data search with different predicted libraries built on public databases is below 5% and comparable to the FLR control with the experimental DDA library. Taken together, we demonstrated the application of an iterative search considerably promotes DIA profiling of the quantifiable phosphoproteome while not inflating the FDR or FLR in data mining.

Based on our DIA quantification results from the iterative search, we performed an ANOVA statistical test to identify significantly regulated phosphosites at EGF or any kinase inhibitor treatment. In concordance with higher coverages, DIA data analysis with any of the five DeepPhospho predicted libraries yielded more regulated sites than Lib 1, with Lib 6 and Lib 7 gaining additional 235 and 212 sites relative to Lib 1 (Fig. 5a). To address one of the central biological questions of this study, we performed a Tukey's range test to identify EFG-regulated sites (significantly changed at EGF treatment *vs* control) with each library which were further divided into MEK-dependent (changes at EGF treatment reversed by inhibitor treatment) and MEK-independent sites (changes at EGF treatment unaffected by inhibitor treatment). All regulated sites uncovered by different libraries are summarized in Supplementary Data 3. Again, all DeepPhospho predicted libraries outperformed Lib 1 in regard to the number of functionally regulated sites in each category (Fig. 5b). For example, data mining with Lib 6 and Lib 7 uncovered 128 and 122 new EGF-regulated phosphosites that were not revealed with Lib 1 (Fig. 5c). Comparison of these new regulated sites with published EGF signaling proteomics results further corroborated the regulation of site-specific phosphorylation uncovered with the predicted libraries (Supplementary Fig. 10).

Next, we performed bioinformatics analysis based on the regulated sites to assess how much biological insights into the phosphosignaling network can be gained by data mining with DeepPhospho predicted libraries. For two libraries of our most

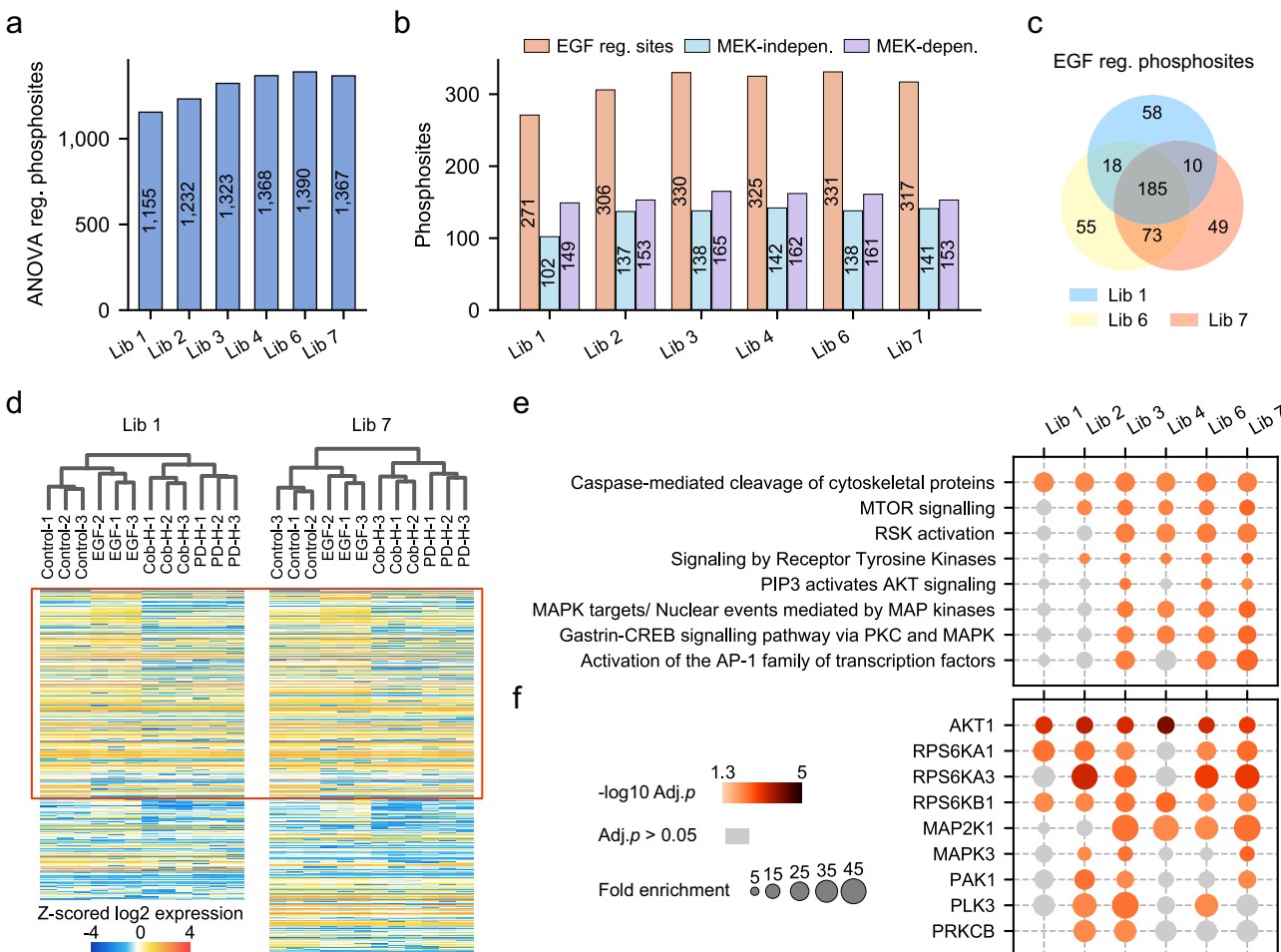

**Fig. 5 DIA phosphoproteomics with DeepPhospho predicted libraries provides more biological insights than an extensive project-specific DDA library.**
**a** More significantly regulated phosphosites (ANOVA $p < 0.05$) were found in the RPE1 cell signaling study by DIA data analysis with five predicted libraries (Lib 2 to Lib 7) than with the DDA library (Lib 1). This DDA library consisted of 89,416 phosphopeptides identified from 147 DDA runs. **b** An increased number of EFG-regulated sites and MEK-dependent or -independent sites were found with predicted libraries compared to Lib 1. **c** A number of new EGF-regulated phosphosites were found with predicted libraries (Lib 6 and Lib 7) compared to Lib 1. **d** Unsupervised hierarchical clustering of significantly regulated phosphosites identified at different stimulation conditions with Lib 1 or Lib 7. The red rectangle indicates phosphosites co-identified by two libraries. **e**, **f** Enriched Reactome pathways (**e**) and over-represented kinases (**f**) based on phosphoproteome profiling results yielded with each library. Fisher's exact test (two-sided) was performed and adjusted by the Benjamini–Hochberg procedure. Significantly enriched pathways or kinases (adjusted $p < 0.05$) are annotated in a color gradient, and enrichment terms with adjusted $p > 0.05$ are shown in light gray. Source data for this figure are provided as a Source data file.

interest (Lib 6 and Lib 7), unsupervised hierarchical clustering of the ANOVA significant sites identified by each was performed in parallel to Lib 1, which revealed very similar patterns of regulation among all stimulation conditions when comparing Lib 7 with Lib 1 (Fig. 5d) or Lib 6 with Lib 1 (Supplementary Fig. 7f). Interestingly, signaling pathway analysis based on EGF-regulated sites revealed 7 additional pathways to be significantly enriched by results from Lib 6 and Lib 7 than from Lib 1 which only enriched one pathway (Fig. 5e). The additional pathways only enriched by DeepPhospho predicted libraries included mTOR, AKT, PKC, and MAPK pathways which are well known signaling axes activated by EGF. Consistently, nine regulated kinases including AKT1, RPS6K, MAPK/MAP2K, and PAK1 were significantly over-represented in results from DeepPhospho predicted libraries in contrast to three kinases over-represented in the result from Lib 1 by the kinase-substrate pair enrichment analysis (Fig. 5f). In summary, bioinformatics analysis of EGF-regulated sites uncovered by DeepPhospho predicted libraries recapitulated the known EGF signaling network to a much larger extent than the project-specific DDA library.

**Performance of DeepPhospho predicted libraries in a quantitative two-proteome model.** The quality of large-scale phosphoproteomic studies depends on not only the proteome coverage but also the quantification accuracy and reproducibility. To evaluate the quantification performance of DIA analysis with our predicted libraries, we used another published dataset acquired from a standard two-proteome model[3] (Supplementary Data 1). In this model, phosphopeptides enriched from yeast were diluted at different ratios into a fixed background of HeLa phosphopeptides, and the mixed phosphoproteome samples at five serial dilution conditions were individually subjected to DIA data acquisition each in six injection replicates (30 DIA runs in total) (Fig. 6a). As a result, the expected ratios of yeast phosphopeptides at four conditions relative to the control would be 0.25:1, 0.5:1, 1.5:1, and 2:1 while human phosphopeptides are expected to have no changes at any condition. As usual, the previous study built an extremely extensive project-specific DDA library consisting of 119,171 phosphopeptide identifications by acquiring 203 runs of DDA data from yeast or human pre-fractionated phosphopeptide samples[3]. We trained DeepPhospho

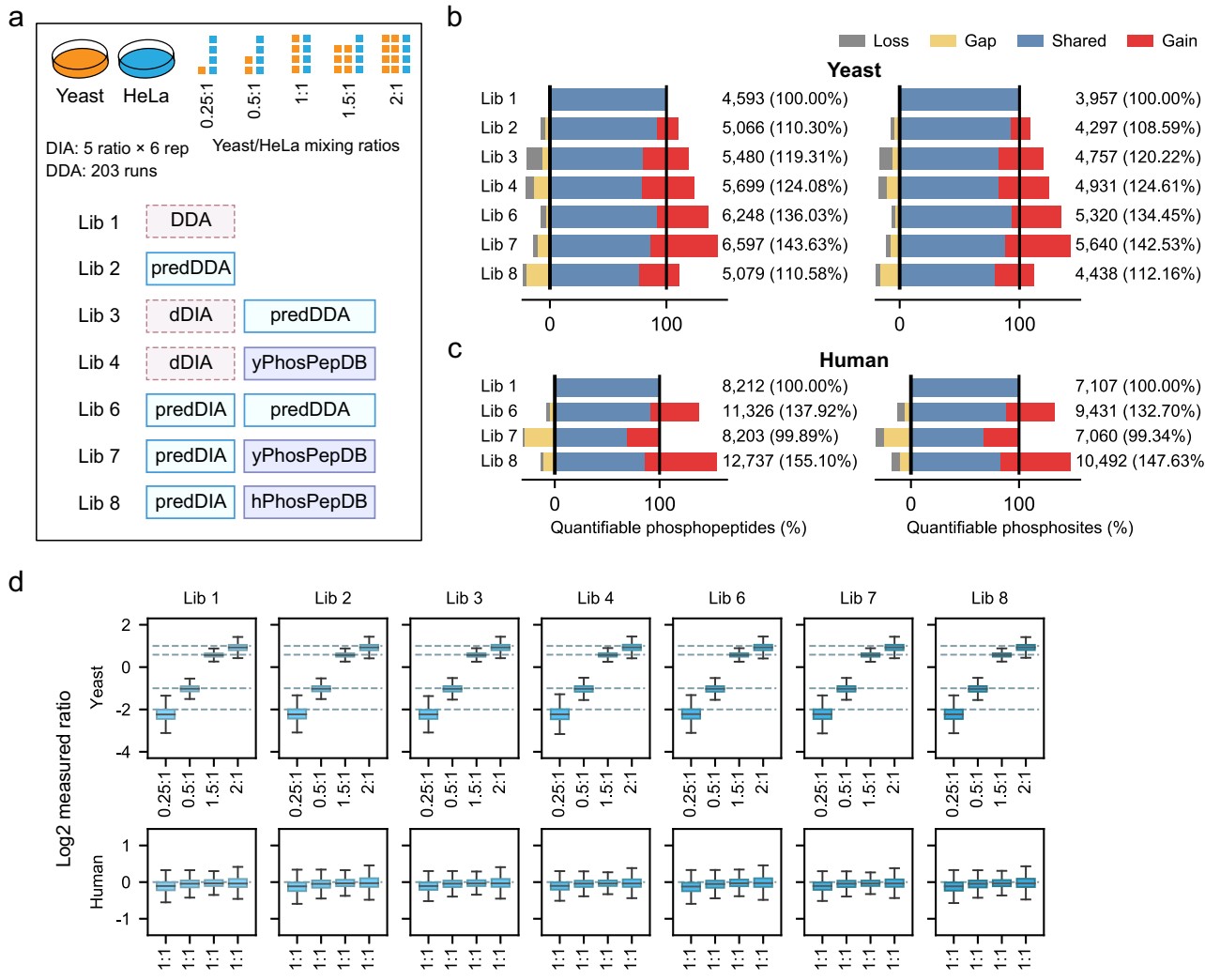

**Fig. 6 Accurate quantification of the phosphoproteome in a two-proteome model. a** Experimental design of the yeast/human two-proteome model and generation of the project-specific DDA library (Lib 1) and six DeepPhospho predicted libraries (Lib 2 to Lib 8). yPhosPepDB, a predicted library built from a previous yeast phosphoproteomic dataset. Other library abbreviations are defined in Fig. 3a. **b, c** Number of phosphopeptides and phosphosites from the yeast (**b**) and human (**c**) proteome that were quantified from the iterative search using each library. Percentage of the total quantifiable phosphopeptide or phosphosite number is shown for each predicted library relative to Lib 1. The proportions of shared identifications (IDs), gained IDs, lost IDs, and gap IDs yielded by Lib 2 to Lib 7 are indicated in different color compared to Lib 1. **d** Boxplots of measured ratios for yeast phosphopeptides (upper) and human phosphopeptides (lower) at different dilution conditions determined by DIA data analysis with each library. Ratios were calculated based on the mean quantities in six replicates of each sample. Expected ratios are indicated by dashed lines and in the x-axis. Boxplot center line, median; box limits, upper and lower quartiles; whiskers, 1.5× interquartile range. Source data for this figure are provided as a Source data file.

with the two-proteome DIA data to create six predicted or hybrid libraries based on predictions for phosphopeptides in the DDA library, direct DIA library, or phosphopeptide sequences from two different resources (Fig. 6a, Supplementary Fig. 11a). Driven by a major attempt to quantify the yeast phosphoproteome with a maximal coverage, we constructed Lib 4 and Lib 7 based on predictions for 36,954 yeast phosphopeptides reported in a deep yeast phosphoproteomic study using various extraction and enrichment approaches[6] (yPhosPepDB) (Supplementary Data 2). Meanwhile, Lib 8 was built to mainly contain a predicted library for human phosphopeptides registered in hPhosPepDB.

Iterative search of the two-proteome DIA data with four DeepPhospho predicted libraries (Lib 3, Lib 4, Lib 6, and Lib 7) resulted in over 20% increase of quantifiable yeast phosphopeptides and phosphosites relative to Lib 1 (Fig. 6b). Remarkably, compared to 4593 phosphopeptides and 3957 phosphosites quantified with Lib 1, Lib 6, and Lib 7 yielded quantifications of 6248 and 6597 phosphopeptides corresponding to 5320 and

5640 phosphosites respectively, both achieving more than 40% increase of coverage. This result suggested the predicted library built on a published phosphoproteomics dataset for the same species in Lib 7 performed very closely to the predicted large-scale DDA library in Lib 6. As a control, the predicted library built on a public human phosphoproteome database in Lib 8 failed to significantly increase the coverage of the yeast phosphoproteome (Fig. 6b). On the other hand, when comparing the coverage of the quantifiable human phosphoproteme in the two-proteome model, we observed 55.1% increase of phosphopeptides and 47.6% increase of phosphosites with Lib 8 relative to Lib 1, whereas Lib 7 showed no increase (Fig. 6c).

This two-proteome model allowed us to precisely assess the quantification accuracy for yeast phosphopeptides serially diluted into a complex phosphoproteome background. DIA data analysis with each DeepPhospho predicted library from Lib 2 to Lib 8 yielded as accurate ratio measurement as Lib 1, with their medians of measured ratios at four dilution conditions very close

to the theoretical values (Fig. 6d upper). Furthermore, our assessment of the quantifiable population of human phospho-peptides as a fixed background revealed equivalent accuracy achieved by each predicted library and Lib 1, with their medians of measured ratios at four dilution conditions all around 1:1 (Fig. 6d lower). In addition, box plot analysis of relative errors between measured and expected ratios demonstrated equally sufficient quantification accuracy of both yeast and human phosphoproteomes across all tested libraries (median per cent relative errors of 7.56–8.18% for predicted libraries and of 7.82% for Lib 1) (Supplementary Fig. 11b). We further evaluated the false rate of phosphopeptide quantification (FQR) with experimental *vs* predicted libraries in the two-proteome model. The FQR curves closely overlapped between Lib 1 and any predicted library from Lib 2 to Lib 8 (Supplementary Fig. 11c). Specific FQRs for yeast and human phosphopeptides at two quantification error thresholds (<50% or <30%) were in general similar among all test libraries (Supplementary Fig. 11d). The equivalent FQRs between our predicted libraries and the benchmark DDA library provided additional evidence of the sufficient FDR control, given that an increase of falsely identified phosphopep-tides would probably result in inflated FQRs. Finally, quantification reproducibility of phosphopeptides between replicates was highly comparable across all test libraries (median CVs of 9.5%-10.1% for predicted libraries and of 9.6% for Lib 1) (Supplementary Fig. 11e). In summary, the high accuracy and reproducibility of DIA quantification results from DeepPhospho predicted libraries underscores their excellent performance not only in quantification but also in identification of the mixed phosphoproteome.

## Discussion

In this study, we present a hybrid deep neural network Deep-Phospho which conceptually differs from all previous deep learning models for unmodified or modified peptide predictions in regard to peptide representation learning. Our approach uti-lizes a multi-module network and self-attention mechanism to learn a highly expressive peptide representation, yielding more accurate predictions. When evaluated with multiple phospho-proteomics datasets acquired by DIA or DDA methods, Deep-Phospho surpasses existing benchmarks and tools in the prediction of fragmentation patterns for phosphopeptides. In certain cases, the large variance between a DeepPhospho pre-dicted MSMS spectrum and an experimentally assigned spectrum revealed the latter was a false identification while the predicted spectrum closely mimics the bona fide spectrum. Moreover, accurate prediction of chromatographic retention time for any phosphopeptide sequence is integrated into DeepPhospho, which allows for convenient construction of in silico spectral libraries to enhance DIA phosphoproteomics data mining.

Transfer learning is a powerful approach to train deep neural networks to learn specific features of the experimental conditions under which a proteomics dataset was acquired[19]. To generate an in silico library suitable for DIA data analysis, researchers chose to train a model such as Prosit and pDeep using data from a DDA experiment which was performed under nearly identical condi-tions to the DIA experiment[11,19]. Unlike previous studies, we trained DeepPhospho using the exact DIA data to be analyzed and showed the transfer learning model afforded high accuracy in prediction of fragment ion intensity and retention time for phosphopeptides in three separate datasets. In principle, a DIA data-trained model can precisely capture the DIA experiment-related parameters that determined the data structure. These parameters would reflect specific LC and MS conditions that typically shift more or less in DDA experiments performed in the

same lab on the same instrument due to internal variation of instruments and the regular change of nanoLC columns. There-fore, an in silico library generated through predictions with a DIA data-trained model is expected to perfectly match the DIA data to be analyzed. As a result, Lib 6, a combination of two predicted libraries converted from a project-specific DDA library and a direct DIA library, enabled a substantial increase of the phos-phoproteome coverage in all three datasets compared to the original experimental libraries. Notably, we used an experimental DDA library as the benchmark here to be compared with dif-ferent predicted libraries. When we built a larger experimental library by merging the direct DIA and DDA libraries, the advantage of predicted libraries to increase the phosphoproteome coverage still remained in all datasets evaluated in this study (Supplementary Fig. 12).

In our study, we designed and evaluated DeepPhospho pre-dicted libraries built on phosphopeptide identifications not only from the experimental DDA or DIA libraries but also from community resources (Fig. 3a). One evident advantage of training the model with DIA data alone and building a predicted library based on public data resources is no need to perform laborious and time-consuming DDA experiments, which could be a pro-nounced improvement of the current DIA workflow. However, it has been recognized that in silico libraries built on public data-bases especially from proteome-scale prediction face a big chal-lenge of extensive query space, which would cause reduced detection sensitivity and increased false positives[14]. Indeed, Lib 5 generated by whole-proteome computation of phosphopeptides based on a human phosphosite database had the largest library size yet the lowest identification rate. In contrast, Lib 4 and Lib 7 both comprised of a predicted library built on phosphopeptides recorded in a human phosphoproteome database substantially expanded the human phosphoproteome coverages in three stu-dies without compromising the FDR control. Most importantly, data analysis with Lib 7 yields a proteome coverage comparable to or even higher than Lib 6. Thus, our study established a DIA workflow for human phosphoproteomics which circumvents the need of DDA experiments and reaches a maximal proteome coverage largely exceeding the state-of-the-art DDA library.

In a classical EGF signaling study, we further demonstrated iterative data search with the best-performing predicted libraries (Lib 6 and Lib 7) enhanced phosphoproteome profiling to a much greater depth (21.2% average increase at the phosphopeptide level and 17.4% average increase at the phosphosite level) than a high-quality extensive DDA library. Of note, we undertook an iterative search strategy to reduce the query space for large-size predicted libraries so as to increase the detection sensitivity and deepen the proteome coverage. Meanwhile, FDR control, quantification accuracy and reproducibility for data analysis with predicted libraries remained as good as, or even better than, the DDA library. Remarkably, more regulated phosphosites were identified with Lib 6 and Lib 7, which led to the significant enrichment of a higher number of EGF signaling pathways and activated kinases than the DDA library. This has major implications that more biological insights could be obtained from DIA phosphopro-teomics analysis if applying our data mining workflow empow-ered by DeepPhospho.

DeepPhospho is provided as a web server (http://shuilab.ihuman.shanghaitech.edu.cn/DeepPhospho) as well as an offline app to facilitate user access to model training, predictions, retention time calibration, and library generation. The ability of DeepPhospho to make high-quality predictions for phosphopeptides enables a DIA phosphoproteomics workflow, in which only single-shot DIA data is acquired for specific samples and data mining completely relies on the DIA data itself and a public database without the need of a project-specific DDA library. For human phosphoproteomics studies,

we provided a complete phosphopeptide input table (hPhosPepDB in Supplementary Data 2) to build an in silico library which can be exploited in other projects. It is noted that the best parameter combination for the hPhosPepDB library differs from the U2OS and RPE1 DIA datasets (Supplementary Figs. 4, 6), highlighting the importance of library parameter optimization. In addition, it awaits further investigation whether usage of the hPhosPepDB library could deepen the DIA phosphoproteome profiling for any samples from human tissues, body fluids, and cell lines.

Given the unique architecture and high performance of DeepPhospho, we envision it can be modified to make accurate predictions for non-phosphopeptides as well as peptides of diverse modifications so as to build efficient DIA workflows for global proteomics and PTM proteomics. Undoubtedly, such workflows would accelerate current proteomics research by enhancing protein/peptide detection as well as reducing sample size and instrument time investment. In addition, we anticipate DeepPhospho to be readily applied to the validation of phosphopeptide identifications, targeted MS assay development for selected phosphopeptides in a complex background, as well as independent assessment of FDR control in DIA analysis. We believe DeepPhospho and our DIA phosphoproteomics workflow would benefit proteomics and biological research in various ways.

## Methods

**Processing of external DDA/DIA MS data.** For initial evaluation of the model architecture, the mouse brain DDA data[29] was downloaded from PRIDE with the identifier PXD006637 and its MaxQuant search output file was directly used. The yeast R2P2 DDA data[6] was downloaded from PRIDE with the identifier PXD013453, and the raw data were searched against the Uniprot *S. cerevisiae* reference proteome (7,500 protein sequences, downloaded in 2020/09) with MaxQuant[30]. MaxQuant v1.6.14.0 was used in this work with the following settings: Phospho (STY), Oxidation (M), and Acetyl (Protein N-term) were set as variable modifications; Carbamidomethyl (C) was set as fixed modification; tolerance of first search and main search were 20 p.p.m. and 4.5 p.p.m.; FDR at PSM level and protein level was set to 0.01; min Andromeda score of modified peptides was 40. The yeast R2P2 DDA data was also used to evaluate the iRT prediction model.

For model pre-training, the mouse brain DDA data mentioned above was used again, together with three other datasets: Vero E6 DIA data[8] (downloaded from PRIDE with the identifier PXD019113), yeast DIA data[6] (downloaded from PRIDE with the identifier PXD013453), and the human phosphoproteome RT data downloaded from the supplementary data of a published work[24] (removing phosphopeptides with the phosphosite Ascore ≤13). Both DIA data were used to build direct DIA libraries using the Pulsar search engine in Spectronaut[31] v14.5 by searching against Uniprot *C. sabaeus* reference proteome (19,136 protein sequences, downloaded in 2020/04) or Uniprot *S. cerevisiae* reference proteome (7500 protein sequences, downloaded in 2020/09). The procedure of building direct DIA libraries is descried in the session of "Spectral library generation".

For evaluation of model prediction for phosphopeptides, we downloaded RPE1 DDA and RPE1 DIA data[3] from PRIDE with the identifier PXD014525 and U2OS DIA data[32] from PRIDE with the identifier PXD017476. RPE1 DDA data initially downloaded in a Spectronaut specific.kit library format was transformed to a plain text file. Then we removed peptide entries with modifications of Deamidation (NQ) and Gln->pyro-Glu which rarely occur and are not supported in current DeepPhospho models. RPE1 DIA data and U2OS DIA data were both searched with Pulsar to generate direct DIA libraries, and Uniport human reference proteome (UP000005640, 84,823 protein sequences, downloaded in 2020/06) was used as the sequence database. Reference spectra of phosphopeptides were obtained from two DDA-based human phosphoproteomic studies: U2OS DDA data[32] downloaded from PRIDE with the identifier PXD017476 and U-87 DDA data[1] downloaded from PRIDE with the identifier PXD009227.

For evaluation of DeepPhospho predicted libraries, we used U2OS DDA and DIA data, RPE1 DDA and DIA data as described above, as well as DDA and DIA data from a human/yeast two-proteome model[3] downloaded from PRIDE with the identifier PXD014525. The yeast DDA library built from the human/yeast two-proteome model data was also provided in a.kit format and processed the same way as RPE1 DDA data. Direct DIA libraries were generated from RPE1 DIA data and U2OS DIA data as described above. In addition, the human/yeast direct DIA library was generated by Spectronaut using the Uniprot human reference proteome and Uniprot *S. cerevisiae* reference proteome as the sequence databases.

All phoshopeptides from the external datasets that were used for model training and evaluation need to have a phosphosite localization score >0.75 in MaxQuant or Spectronaut output files (class I sites). Details in the external data source, sample

source, MS instrument condition, and data processing are described in Supplementary Data 1.

**Processing of the phosphoproteome and phosphosite databases.** To construct Lib 4, Lib 5, and Lib 7 in Fig. 3, Lib 4 and Lib 7 in Fig. 4, Lib 4, Lib 7, and Lib 8 in Fig. 6, we created three databases for generation of predicted libraries: hPhosPepDB, hPhosSiteDB, and yPhosPepDB. We built hPhosPepDB based on a published human phosphoproteome database[24] which recorded the sequences, PTM sites, charge states and calibrated RT for 204,606 label-free, trypsinized, confidently localized phosphopeptides (Ascore >13) from 12,228 proteins detected in large-scale phosphoproteomic experiments from various sources. To find out the best condition for generation of a predicted library from hPhosPepDB, we restricted precursor and fragment mass ranges, peptide length, max phosphosite number, and charge state in specific predicted libraries for performance testing. This was performed to generate the optimized Lib 4 and Lib 7 in both Figs. 3 and 4, and Lib 8 in Fig. 6.

We then built hPhosSiteDB based on the human protein phosphosites registered in EPSD database[25] and in silico digestion of the whole human proteome. Specifically, in silico phosphopeptide sequences were computed using these criteria: trypsin specificity in digestion; peptide length from 7 to 30; no miss cleavage; adding phosphosites that are registered for specific proteins in EPSD; max phosphosite number in each peptide is 1. As a result, hPhosSiteDB contained 350,719 unique phosphopeptide sequences. Their charge states were defined as 2, 3 and 4.

The yeast phosphoproteme database yPhosPepDB was built based on 36,954 yeast phosphopeptides detected in a yeast R2P2 phosphoproteomic study using various extraction and enrichment approaches[6]. The original charge states assigned in MaxQuant output files are kept for all phosphopeptides in yPhosPepDB.

More details of the three databases are provided in Supplementary Data 2 and their data sources are listed in Supplementary Data 1.

### DeepPhospho model

*Notations and data representation.* Each input peptide is represented by a sequence of amino acid tokens denoted as L, K, M, etc., typically 7–50 in length. For phosphopeptides, we use 1 to represent the oxidation of methionine (M), and 2, 3, 4 to represent the phosphorylation of serine (S), threonine (T), tyrosine (Y), respectively. In addition, DeepPhospho supports peptides with an N-terminal acetyl modification. We use the * symbol to indicate modification and @ to indicate no modification.

For the task of fragment ion intensity prediction, we denote the model input as $X_F := [x_0, x_1, x_2, \cdots, x_n, +q]$, where $x_0$ is the token of * or @, $x_i (i \geq 1)$ denotes the amino acids, $n$ is the peptide length, and $+q$ is the peptide precursor charge. The output spectrum or the peptide fragmentation pattern $y$ is represented by a matrix of size $L \times 8$ where $L$ is the maximum peptide length in the dataset and each row is a set of intensity values for different combinations of b/y ions, two charge states (+1 or +2) and with or without loss of phosphate (-1,H3PO4 or -noloss). Concretely, the i-th row of the pattern represents the intensities of the following fragments: bi+1-noloss; bi+2-noloss; bi+1-1,H₃PO₄; bi+2-1,H₃PO₄; yi+1-noloss; yi+2-noloss; yi+1-1,H₃PO₄ and yi+2-1,H₃PO₄. Fragment ion intensity values at impossible dimensions are set to −1 while the rest are normalized to [0, 1].

For the task of iRT prediction, we use the peptide without charges as our input, denoted as $X_R := [x_0, x_1, x_2, \cdots, x_n]$, where $x_0$ and $x_i (i \geq 1)$ are described above. The output z is the retention time and is normalized to [0, 1] for each dataset.

*Model architecture.* The DeepPhospho model consists of three main modules, including an embedding network, a sequence modeling network and a regression network. The embedding network encodes the input tokens into feature vectors while the regression network generates output predictions. As the fragment ion intensity and iRT prediction have different forms of input and output, we adopted separate designs for the embedding and regression network in those two tasks.

(1) Embedding network. For the fragment ion intensity prediction, we first embed each amino acid and the charge to vectors of 192 and 64 dimensions respectively and then concatenate them as inputs to the sequence modeling module. For the RT prediction, we directly embed each amino acid into a vector of 256 dimensions.

(2) Sequence modeling network. We adopt a hybrid network for the main module of our model, which consists of a bidirectional Long Short-Term Memory (biLSTM)[33] subnet and a Transformer[34] subnet. Our biLSTM subnet comprises two stacks of bidirectional LSTM with hidden dimensions of 512. This module aims to compute an initial representation of the peptide sequence, which is then fed into the second module, the Transformer subnet. The Transformer aims to capture long-range dependency in the peptide sequences with more effective attention mechanism. Our Transformer subnet stacks multiple Transformer encoders, each of which has 8 self-attention head. We also use the standard sine and cosine functions as the position encoding[34]. More specifically, for the fragment ion intensity prediction, the Transformer subnet comprises 8 layers of Transformer encoders. For the iRT prediction, we use an ensemble of networks with 4 to 8 layers of Transformer encoders.

(3) Regression network. We use a simple linear layer to project the features at each amino acid site to a vector of 8 dimensions as our output in the task of fragment ion intensity prediction. For the iRT prediction, we introduce a linear layer to generate an instance-specific weight for sequence features and use a weighted average to produce the RT prediction.

*Model training.* We adopted a transfer learning strategy to train our models. For the task of fragment ion intensity prediction, we first pre-trained our model on three datasets (Supplementary Data 1), which provided a good initialization (Supplementary Fig. 13). We then fine-tuned the pre-trained model on each of three target datasets (Supplementary Data 1). Specifically, for the pre-training datasets, we split each of them into a training and a validation set with a 9:1 ratio; for the three target datasets, we split each of them into training, validation, and test set with an 8:1:1 ratio. We used the mean squared error (MSE) as our loss function and the Adam update[35] to optimize the loss with learning rate 1e−3 on the first pre-training phosphoproteome dataset, and 1e−4 on the other datasets. We decayed the learning rate by 0.1 after pre-defined number of epochs. We tuned the model hyper-parameters and selected the best model on the validation set. We reported the result on the test set of the target datasets. For the task of iRT prediction, the model was initially pre-trained with four datasets (Supplementary Data 1, Supplementary Fig. 13), in which the RT values were normalized into [0,1] on all datasets. For the target datasets, we manually set the $min(RT)$ and $max(RT)$ equals −100 and 200, respectively. We used the root mean square error (RMSE) loss and applied the same training strategy as in the fragment ion intensity task. In three DIA phosphoproteomic studies, the pre-trained models were specifically trained with each DIA dataset to generate trained DeepPhospho models for predicted library construction.

The models were trained either on 4 TITAN Xp GPUs or 4 GTX1080Ti GPUs. The training time is related to the size of a dataset. For the fragment ion intensity model, it took about 12 h for pre-training with a dataset containing 80k precursors, and about 2 h for fine-tuning with a dataset containing 40k precursors. For the iRT model, it took about 22 h for pre-training with a dataset containing 180k peptides, and about 4 h for fine-tuning with a dataset containing 30k peptides.

*Metrics.* For the task of fragment ion intensity prediction, we compute the Pearson correlation coefficient (PCC) between the prediction and the ground truth of each peptide and select the median of those PCCs as the final evaluation metric. In addition, we follow Prosit[11] and use normalized spectral angle (SA) as another metric, and also report the median of those SAs. The normalized spectral angle is defined as follows.

$$SA(\hat{y}, y) = 1 - 2 * \frac{cos^{-1}(\hat{y} \cdot y)}{\pi} \quad (1)$$

where $\hat{y}, y$ are two vectors whose L2 norm equals 1. We select the model by the median PCC metric.

For the task of iRT prediction, we adopt the $\Delta t_{95\%}$ metric as the main metric, which represents the minimal time window containing the deviations between observed and predicted RTs for 95% of the peptides:

$$\Delta t_{95\%} = 2 * |z - \hat{z}|_{95\%} \quad (2)$$

The subscript 95% means the 95% rank of the deviations.

*Model architecture validation.* To validate our model design, we conducted a set of ablative study on the model architecture. Specifically, we compared our model with several alterative model designs, including the biLSTM module only, the Transformer module only, and replacing biLSTM with a CNN module (CNN+Transformer). We used a variant of ResNet34[36] in our setting. We performed the comparisons on both the fragment ion intensity and iRT prediction benchmarks (Supplementary Data 1). We split each dataset into training: validation: test = 8:1:1, and after model selection on the validation set, we reported the results on the test set. PCC and SA were used to validate the prediction of fragment ion intensity and median absolute error (MAE) was used to validate the iRT prediction.

*Comparison of DeepPhospho with other models.* We compared our method with several published models, including pDeep2, DeepMS2, and MS2PIP on three datasets: RPE1 DDA, RPE1 DIA, and U2OS DIA. For pDeep2, the source code and pre-trained model parameters pretrain-180921-modloss.ckpt were downloaded from their website https://github.com/pFindStudio/pDeep/tree/master/pDeep2, and transfer learning was performed with the default hyper-parameters. For DeepMS2 (https://github.com/lmsac/DeepDIA), we used the pre-trained model parameters epoch_035.hdf5 to make predictions for precursors with 2+ charges and epoch_034.hdf5 for precursors with 3+ charges. Then we followed the budding strategy described by the authors[20] to generate in silico spectra for phosphopeptides with scripts stored at https://github.com/lmsac/DeepMS2-phospho. For MS2PIP, we directly used the MS2PIP server (https://iomics.ugent.be/ms2pip) and chose the model HCD (including b++ and y++ ions) for prediction.

**Analysis of synthetic phosphopeptides.** Seven phosphopeptides were synthesized by GenScript (Nanjing, China). The peptide powders were dissolved in

ultrapure water or DMSO to prepare 5-10 mg/ml stocks. The stock solution was diluted to 100 ng/μl using 0.1% FA and seven phosphopeptides were mixed together before injection into the nanoLC-MS system for DDA and PRM data acquisition.

The nanoLC-MS/MS analysis was conducted on an EASY-nLC 1200 connected to QE HF mass spectrometer (Thermo Fisher Scientific, USA) with a nano-electrospray ionization source. The peptide mixture of 10 ng was loaded in each replicate and separated on an analytical column (200 mm × 75 μm) in-house packed with C18-AQ 1.9 μm C18 resin (Dr. Maisch, GmbH, Germany) over a 60-min gradient from 4 to 45% mobile phase B (0.1% FA in acetonitrile) at a flow rate of 300 nl/min. In DDA data acquisition, the resolution of Orbitrap analyzer was 60,000 for MS1 and 15,000 for MS2. The AGC target was set to 3e6 in MS1 and 1e5 in MS2, with a maximum ion injection time of 120 ms in both MS1 and MS2. The isolation window was set to 1.6 *m/z*, and stepped collision energy at 25, 27, and 30. In PRM data acquisition, the resolution of Orbitrap analyzer was 120,000 for MS1 and 30,000 for MS2. The AGC target was set to 3e6 in MS1 and 5e5 in MS2, with a maximum ion injection time of 20 ms in MS1 and 120 ms in MS2. The isolation window was set to 1.0 *m/z*, and stepped collision energy at 25, 27, and 30. The inclusion list contained the precursor m/z and RT windows for the seven phosphopeptides that were detected in the DDA experiment.

Acquired DDA raw data was analyzed using MaxQuant (v1.6.17.0) against the seven phosphopeptide sequences appended with a contaminant sequence database. The following search parameters were used: no fixed modification, Phospho (STY) as variable modification, and trypsin as specific enzyme. The first search tolerance was set to 20 ppm, main search tolerance to 4.5 ppm, filtered for PSM and protein FDR of 1%. Then the msms.txt file exported from MaxQuant was imported into Skyline[37] (v20.2.0.343) to build a library. PRM data was analyzed by Skyline with the major settings: precursor charges 2, ion charges 1 and 2, ion types p, b, y, product ion selection from *m/z* > precursor to last ion, library pick product ions 25, use scans within 30 min. All XICs of selected fragments were manually inspected and adjusted to ensure proper peak picking and peak integration. An "idotp" value of each precursor of >0.9 was accepted.

**Spectral library generation.** The project-specific DDA library used in the U2OS data analysis was built from 20 runs of DDA data in Spectronaut (version 14.5, Biognosys AG, Switzerland). The DDA data were imported to Spectronaut to generate the DDA library by Pulsar with default settings except the addition of Phospho (STY) as variable modifications, Best N Fragments per Peptide Max set to 25, PTM Min Localization Threshold set to 0.75, and Fragment ions *m/z* set to 200–2000. The DDA libraries used in the analysis of RPE1 data and the two-proteome model's data were provided by previous work[3] and downloaded from ProteomeXchange with the identifier PXD014525. They were initially built from 147 runs of DDA data for RPE1 project and 203 runs of DDA data for the two-proteome model using Spectronaut (v11.0.15038.19 and v13.0.190309, Biognosys AG, Switzerland) with default settings except the addition of variable modifications as Phospho (STY) and Best N Fragments per Peptide Max set to 25.

Direct DIA libraries used in this work were all built from the raw DIA data and generated with Pulsar in Spectronaut v14.5 with default settings except the followings: Phospho (STY) as variable modifications as, Best N Fragments per Peptide Max set to 25, PTM Min Localization Threshold set to 0.75, and Fragment ions *m/z* set to 200–2000.

For the generation of predicted libraries, a list of phosphopeptide sequences collected from the DDA library, the direct DIA library or a specific phosphoproteme or phosphosite database was input to the trained DeepPhospho models for the prediction of fragment ion intensity and iRT. In some cases, a direct DIA library need to be merged with a predicted library to yield a hybrid library. For any redundant peptides present in both the direct DIA library and the predicted library, their experimental MSMS spectra and iRT values in the direct DIA library were retained in the hybrid library. All predicted libraries and hybrid libraries generated in-house were written in a tab separated value (TSV) file to be processed by Spectronaut in DIA data analysis.

In the iterative search, we created a focused library corresponding to each complete library used in the initial search. For Lib 1 and Lib 2 in Figs. 4 and 6, their focused libraries only contained peptides identified in the initial search. For Lib 3 to Lib 8, their focused libraries were composed of a DIA library and a predicted library. While the predicted libraries only contained peptides identified in the initial search, the experimental DIA library contained all detected peptides.

**DIA data analysis and FDR/FLR assessment.** Raw DIA data were processed using Spectronaut v14.5 with default settings. In brief, PTM localization was activated and site probability score cutoff was set to 0.75, data filtering was set to Q-value and Normalization Strategy set to Global Normalization. Decoy generation was set to mutated. Interference correction was enabled and the number of minimum inferenced ions was 2 and 3 for MS1 and MS2, respectively. Peptide and protein level Q-value cutoff was set to 1%. In each analysis, a specific experimental library, predicted library, or hybrid library generated earlier was imported to Spectronaut. For U2OS DIA data and RPE1 DIA data analysis, Uniprot human reference proteome (UP000005640, 84,823 protein sequences, downloaded in 2020/06) was used as the protein sequence database. For the analysis of the two-proteome model, Uniprot human reference proteome and Uniprot *S. cerevisiae*

reference proteome (7500 protein sequences, downloaded in 2020/09) were used. After DIA data processing, the peptide and protein reports were exported for further statistics and bioinformatics analysis.

Although Spectronaut automatically set FDRs to be <1% at both peptide and proteins levels, we employed three additional methods for assessing the library-specific FDR. First, we created a reverse library for each original library by reversing the sequences of all peptides in the original library except the C-terminal residue, and keeping the original charge states. The reversed peptides were imported to the trained DeepPhospho models for predictions of their fragment ion intensities and iRT values to generate a reverse library which was then appended to the original library. Searching DIA data with this original-reverse combined library provided a rough estimate of FDR for peptide-centric analysis as defined below

$$\text{FDR} = 2 \times \frac{Hits_{reverse}}{Hits_{original} + Hits_{reverse}} \quad (3)$$

Second, we created two-species libraries by combining a predicted human phosphoproteome library and a predicted *A. thaliana* phosphoproteome library. This predicted *A. thaliana* library was generated by predictions for 103,620 precursors corresponding to 34,540 unique phosphopeptide sequences registered in PhosPhat database which compiles various DDA-based phosphoproteomics data[38]. Searching DIA data with a two-species library yielded the estimated FDR defined as

$$\text{NormHits}_{False} = \text{Hits}_{False} \times \frac{LibSize_{Human}}{LibSize_{A.thaliana}} \times \frac{1}{1 - 0.01} \quad (4)$$

$$\text{FDR} = \frac{NormHits_{False}}{Hits_{True} + NormHits_{False}} \quad (5)$$

where $LibSize_{Human}$ is the number of all phosphopeptides in the human sub-library, and $LibSize_{A.thaliana}$ is the number of all phosphopeptides in the *A. thaliana* sub-library. The constant item $\frac{1}{1-0.01}$ is multiplied to offset the inherent error tolerance of 1% FDR for all results. Notably, it is difficult to calculate an accurate FDR using the entrapment strategy given that it is unlikely to know the exact number of phosphopeptides in the human sub-library that is not present in the sample. This metric is provided to roughly estimate and compare the error rates of phosphopeptide identification when using different libraries.

Last, we calculated the false rate of phosphopeptide quantification (FQR) under varying thresholds of the quantification error between measured and theoretical ratios of phosphopeptides identified at all dilution conditions with different libraries in the two-proteome model. Although the absolute value of FQR cannot represent that of FDR, its relative comparison under different conditions could serve as a metric to evaluate whether the FDR of DIA data analysis with any predicted library is inflated compared to the benchmark DDA library.

Furthermore, we employed a classical approach to estimate the false localization rate (FLR) of phosphosites using a DIA dataset acquired on 166 synthetic human phosphopeptides containing 176 clearly defined phosphosites[27]. We first generated experimental DDA libraries comprised of the synthetic phosphopeptide data alone (SynLib) or appended with an extensive human phosphoproteome library (RPE1 DDA). We also generated a predicted library based on the synthetic phosphopeptide information (predSynLib) and constructed hybrid libraries by appending predSynLib with a large predicted library built on hPhosSiteDB or hPhosPepDB. Initial and iterative searches were performed for all hybrid libraries combining the synthetic phosphopeptide library and a much larger experimental or predicted library. FLR was calculated based on the false and total phosphosites identified in the synthetic peptide set.

**Statistics and bioinformatics analysis.** In the phosphosignaling study with RPE1 DIA data, Spectronaut reports were first modified to be compatible with PerseusR[39], and then transformed into a modification specific peptide-like reports using Peptide Collapse[3], with the target PTM site as the collapse level, localization cutoff 0.75, and variable PTMs in the order of Phospho (STY), Oxidation (M) and Carbamidomethyl (C). The reported intensities were log2-transformed and z-scored. Quantifiable phosphopeptides and phosphosites were selected if their intensities were measured in all three replicates for at least two different treatments. One-way ANOVA test was applied to the quantifiable phosphosites to identify significantly regulated sites ($p < 0.05$) at EFG or any kinase inhibitor treatment *vs* control (Fig. 5a). The Tukey's range test implemented in statsmodels was then applied to ANOVA-significant phosphosites to identify the EGF-regulated sites which were significantly changed at EGF treatment *vs* control (adjusted $p < 0.05$). The EGF-regulated sites were further divided into two classes: one is the MEK-dependent phosphosite which was also significantly changed according to the Tukey's range test at one of the kinase inhibitor treatment with the opposite trend of regulation to the EGF treatment; the other is the MEK-independent phosphosite which showed no significant regulation at any kinase inhibitor treatment (Fig. 5b).

The hierarchical clustering was implemented to ANOVA-significant phosphosites identified at EGF or high-dose inhibitor treatment using Scipy[40], with the metric set to correlation and the method set to average. To fill in the expression matrix for clustering and heatmap, NA values were imputed by randomly sampling values from a normal distribution with the mean of −1.5 and standard deviation of 0.5. Heatmaps of all significantly regulated phosphosites were generated by unsupervised hierarchical clustering.

The kinase-substrate pair enrichment was performed based on the kinase-substrate relationship downloaded from PhosphoSitePlus[41] (access date: 2021/01) using the fisher exact test implemented in Scipy[40]. We used all EGF-regulated phosphosites as the input and all identified phosphosites as the background. Signaling pathway enrichment was then performed based on the Reactome pathway data[42] (access date: 2021/01) using the fisher exact test. EGF-regulated phosphosites were first collapsed to the protein level and used as the input while the background was all identified phosphoproteins. A total of 14 significantly enriched pathways and 13 significantly enriched kinases (adjusted $p < 0.05$) were initially identified using any of the six tested libraries. Eight enriched non-redundant pathways and nine over-represented kinases that were discovered using at least two spectral libraries were kept and shown in Fig. 5e, f.

In the quantitative two-proteome model study, the Spectronaut exported peptide precursor intensities were first de-normalized by dividing reported intensity values by their normalization factors. The measured ratio of a phosphopeptide identified at any of the four dilution conditions (0.25:1, 0.5:1, 1.5:1, and 2:1) were calculated by dividing its intensity measured at that condition by the intensity measured at the control condition (1:1). Then quantifiable yeast or human phosphopeptides and phosphosites were selected if they had at least one ratio measured at any dilution condition relative to control.

Boxplots were created with boxes marking the first and third quartile, a dash the median, and whiskers the minimum/maximum value within 1.5 interquartile range. Outliers are not displayed.

Data analysis in this part was performed using python (3.7.9) and the following packages: numpy (1.19.2), scipy (1.6.0), pandas (1.2.1), and statsmodels (0.12.0). Visualization was achieved with matplotlib (3.3.2), matplotlib-venn (0.11.5), and seaborn (0.10.1).

**DeepPhospho web server and offline app.** We used the open-source web framework Flask and frontend framework Vue.JS for developing the web based live demo. In the START page, users can make predictions of MSMS spectra and iRT values for either a single phosphopeptide or a batch of phosphopeptides with defined sequences and charge states. In the batch mode, after inputting the phosphopeptide information, users will be able to download a.txt file as a ready-to-use spectral library for DIA data mining. In this web server we provide four DeepPhospho models trained with specific DDA/DIA MS datasets that were acquired from different sample sources and under different LC-MS/MS settings. These trained models can make accurate predictions for phosphopeptides analyzed under similar instrument conditions. We also provide an option for iRT calibration so that the predicted iRT can fit into the experimental RT scale defined by the user.

For the analysis of data acquired at distinct conditions, we have created an offline DeepPhospho app for users who need to do transfer learning with their own datasets. This offline app allows users to directly use the pre-trained model, train a new model, or fine-tune the model parameters with their own target datasets before making predictions with a selected model. Using our offline DeepPhospho app, a ready-to-use predicted spectral library will be generated as an output file. The offline app can be downloaded from GitHub repository. Alternatively, users can download and explore our user-friendly DeepPhospho pipeline stored in GitHub repository.

**Reporting summary.** Further information on research design is available in the Nature Research Reporting Summary linked to this article.

## Data availability
Raw DDA and PRM data from synthetic phosphopeptide analysis, DeepPhospho generated spectral libraries, and DIA search results have been deposited to the ProteomeXchange Consortium[43] via the iProX[44] partner repository with the dataset identifier IPX0003513000, which is equivalent to PXD028601. Public MS data used in this work are as follows: PXD006637 (mouse brain DDA)[29], PXD019113 (Vero E6 DIA)[8], PXD013453 (yeast R2P2)[6], PXD014525 (RPE1, two-proteome)[3], PXD017476 (U2OS)[9], PXD009227 (U-87 DDA)[1], PXD019797 (human synthetic phosphopeptide dataset)[27], PXD004573 (yeast synthetic phosphopeptide dataset)[28]. All MS raw data were downloaded from PRIDE FTP site via FileZilla (v3.51.0) or from jPOST via Mozilla Firefox. Databases used in this work are: UniProt[45] (https://www.uniprot.org), EPSD[25] (http://epsd.biocuckoo.cn), PhosphoSitePlus[41] (https://www.phosphosite.org/staticDownloads), PhosPhAt[38] (http://phosphat.uni-hohenheim.de), Reactome[42] (https://reactome.org). Source data are provided with this paper.

## Code availability
DeepPhospho is written in Python and uses PyTorch to implement deep neural networks. The source code, documents, and related scripts are stored on GitHub (https://github.com/weizhenFrank/DeepPhospho) and Zenodo (https://doi.org/10.5281/zenodo.5594736)[46]. The DeepPhospho web server is available at http://shuilab.ihuman.shanghaitech.edu.cn/DeepPhospho.

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

## Acknowledgements

We are grateful to Dr. Hao Chi, Dr. Wenfeng Zeng, and Dr. Liang Qiao for fruitful discussion on FDR and FLR assessment. This work was funded by ShanghaiTech University, the National Program on Key Basic Research Project of China (2018YFA0507004 to W.S.), National Natural Science Foundation of China (31971362, 32171439 to W.S.) and Shanghai Natural Science Foundation Grant (18ZR1425100 to X.H.).

## Author contributions

W.S. and X.H. conceived and supervised the project. R. Lou performed library generation, MS data processing, and bioinformatics. W.L. developed the DeepPhospho model with the help of R. Li and performed model training. R. Li developed the DeepPhospho web server and offline app with the help of R. Lou and W.L.. S.L. performed MS analysis of synthetic peptides. W.S., X.H., R. Lou, and W.L. wrote the manuscript with inputs from all authors.

## Competing interests

The authors declare no competing interests.
