## [Peer Review File · Nature Communications]

REVIEWER COMMENTS

Reviewer #1 (Remarks to the Author):

The manuscript by Lou et al reports DeepPhospho, a deep learning model that can predict MS2 and iRT of phosphopeptides. The developed deep network is different from the existing ones in MS2 and iRT prediction, e.g. pDeep and Protist, using a transformer network for refining the peptide representation, which is recently widely used in the natural language processing. The authors firstly demonstrated the accurate prediction of MS2 and iRT of phosphopeptides, and then applied the method in DIA phosphoproteomics. Different from DDA experiments-based spectra library, the in-silico library was built based on phosphopeptide sequences from directDIA results of the samples and from published papers or public database, with predicted MS2 and iRT. With the in-silico library, the authors showed that the phosphoproteome coverage was expanded, and its promising potential in kinase signaling studies. Despite the appealing concept, there are some inherent defects, i.e. the control of FLR and FDR assessment when using unmatched libraries, which restrict the publication of the work.

- The current version of the manuscript does not address the most significant issue in phosphorylation – localization of phosphosites in peptides. In the current work, the authors simply chose 0.75 localization confidence, referring to the work by Bekker-Jensen et al. Nevertheless, in the referred work, the library was built by directDIA, wherein the localization of phosphosites is controlled during the directDIA analysis in a way similar to typical DDA data analysis. Then the issue of FLR control in the DIA data analysis is simplified. In the current work, by introducing the hPhosSiteDB and the hPhosPepDB, there is high risk of introducing many phosphosites that do not exist in the analyzed sample. And in the case, the 0.75 localization confidence cannot guarantee a 1% FLR as expected. We have actually made some tests, and it was found that the identified phosphosites by DIA analysis highly depended on the information of phosphosites given in the spectral library. When there was a large portion of wrong phosphosites in the spectral library, the FLR from the DIA analysis results increased significantly. Without address the issue of FLR control when using non-sample specific spectral library, the downstream analysis on kinase signaling studies could include many false information. The authors need to assess the FLR using the predicted spectral libraries in comparison to DDA-based libraries.

- The authors used “decoy” libraries with reversed peptides for library-specific FDR assessment. The reversed “decoy” library was appended to the original “target” library. The combined library was then imported into Spectronaut and used as the actual target library in the target-decoy approach performed by Spectronaut. Since the reversed peptides are too similar to the “mutated” decoys generated by Spectronaut, they are more inclined to be filtered by Spectronaut than false peptides in real samples, and therefore the FDR is underestimated. Since the predicted libraries of the hPhosSiteDB and the hPhosPepDB can include many peptides do not exist in the sample, the target-

decoy approach by Spectronaut may not well control the FDR of the DIA analysis results. The authors need to evaluate FDR independently of Spectronaut's decoy method. Also, the reviewer suggests referring to the original and manually added "decoy" libraries as other terms instead of target and decoy to avoid confusion.

- In model training, the pre-train fine-tune pipeline is a bit weird. This pipeline is for multi-task training, while in this case, tasks are the same but on different datasets. Generally, it will be more reasonable if model is pre-trained on the combination of previous pre-trained datasets and target datasets and then finetuned on the target datasets.

- Comparison in Figure 1b is not fair since the DeepPhospho is trained with the data from the same dataset as the test, while the others are not.

- It is not clear how is the performance of MS2 and iRT prediction for single and multi phosphopeptides, and for S, T, Y, separately?

- The hPhosPepDB library was generated based on a phosphopeptide precursor list collected from a mass spectrometry dataset, which contains sequences, PTM sites and charge states. The hPhosSiteDB library was generated based on protein phosphosite list, which only contains sequences and PTM sites, and the library contains precursors with all the defined charge states for each phosphopeptide. The authors should give this information in the main text.

- Before performing analysis using dDIA+predDDA (in Figure 3), the authors need to report the results by dDIA only.

- Community resources of PTM sites are usually more easily accessible and have more complete coverage than mass spectrometry datasets from specific publications. Therefore, besides Lib 7 predDIA+hPhosPepDB (in Figure 3 and 4), the reviewer would like to see the performance of analysis using predDIA+hPhosSiteDB. Also, can iterative search help to resolve the challenge of extensive query space when using libraries generated from protein phosphosite list (Lib 5 and predDIA+hPhosSiteDB)?

- The iterative search expands the fraction of the quantifiable phosphoproteome within the entire profiled proteome. The results suggest that the same peptide can be assigned to different features in the focus search and the general search. Then there is question on which assignment is correct.

- It shows that data mining with Lib 6 and Lib 7 uncovered 128 and 122 novel EGF-regulated phosphosites that were not revealed with Lib 1. Can these newly identified phosphosites be verified, especially with the concern of the lack of FLR control in the current method?

Minor:

- The PCC scores between the spectra in Figure 2c should be indicated.

- It should be stated in the title that deep learning is used for spectral library generation.

- The Spectronaut spectral libraries (.kit) and search results (.sne) files should be provided in the ProteomExchange repository.

Reviewer #2 (Remarks to the Author):

Summary

The authors describe a novel neural network (nn) architecture that allows to predict fragment ion intensities and indexed retention time (iRT) for phosphorylated peptides. The authors show that predictions with this nn outperform published nn. The authors suggest a novel workflow where the nn network is trained (transfer learning) for a setup using a library derived from searching the DIA data directly. The peptides from a database and from the spectrum-centric DIA search are used to defined the precursors to predict. The authors show that with this workflow they can outperform workflows based on DDA libraries while saving extra measurement time that would otherwise be spent on the DDA runs. The research presented is relevant and of sufficient quality for NC. The manuscript is relatively dense and sometimes difficult to follow. My main concerns are listed below under major points. The authors show metrics that were achieved by applying transfer learning to the model presented. However, using the web interface such transfer learning cannot be applied to the model. Hence, the performance that users of the web interface may experience can differ from the one presented in this manuscript. Minimally the authors should show the difference of

performance in the respective figures (see below). Ideally, the web interface would also allow transfer learning by uploading a library.

Major Points

- Fig 1 Please add all metrics also for the purely pre-trained model without the transfer learning. Users of the web interface will not be able to perform the transfer learning and hence will see a performance that is more similar to the pre-trained model.
- page 7, line 161: There are other potential reasons why the empirical and predicted spectra agree better for DDA. For instance collision energy is determined more homogeneously for precursors in DDA as compared to DIA. In DIA collision energies cannot be chosen optimally for all precursors in a precursor selection window. FDR control in spectrum-centric searching is very established and hence FDR can be assumed to be similar between DDA and DIA libraries. However, it is a fair point that ~1% of entries in the libraries are wrong and could be corrected by prediction. I'd suggest to re shape the text in a way that does not imply that FDR in DIA libraries is generally elevated but rather point to the fact that prediction has the advantage that there are no more "wrong" assays. Currently, there is not enough evidence in the manuscript that would suffice to indicate to an elevated FDR (as mentioned above lower SA or PCC can have other reasons). I'd suggest to move everything related to that point into the supplementary.
- The provided webpage does not allow transfer learning. Ideally the web service would also allow transfer learning by uploading a spectral library.
- page 39, line 887: In the spectrum-centric analysis the basic assumption that false identifications will uniformly distribute over target and decoy database provide a simple framework for FDR estimation (Elias and Gygi et al. 2007). This simple basic assumption is not transferrable to the peptide-centric analysis. Please state in the methods text that this formula can only serve as a rough estimate in the case of peptide-centric analysis.

Minor Points

- page 4, line 84: Please better differentiate point 1) from 2)
- page 5, line 93: Please mention the number of parameters of the main model used in the main text and the number of parameters of all models tested in the supplementary. Ideally compare to number of parameters of other models presented in literature and benchmarked in this study.

- page 6, line 134: Please mention the total number of precursors used for the pre-training.
- Please better explain in the methods how peptides with n phosphosites were spanned (col 6, suppl fig 4)
- page 12, line 280: Please also indicate the library type and size in a sub figure similar to fig 3
- page 12, line 283: Please better explain the iterative search and the metrics used. On how many runs was the initial search performed? Were average figures used for the number of phosphopeptides and sites reported (otherwise I wouldn't understand how the numbers can go up in the iterative search strategy)?
- please mention training hardware (GPUs?) and time (hours, epoches) in methods

REVIEWER COMMENTS

Reviewer #1 (Remarks to the Author):

The manuscript by Lou et al reports DeepPhospho, a deep learning model that can predict MS2 and iRT of phosphopeptides. The developed deep network is different from the existing ones in MS2 and iRT prediction, e.g. pDeep and Protist, using a transformer network for refining the peptide representation, which is recently widely used in the natural language processing. The authors firstly demonstrated the accurate prediction of MS2 and iRT of phosphopeptides, and then applied the method in DIA phosphoproteomics. Different from DDA experiments-based spectra library, the in-silico library was built based on phosphopeptide sequences from directDIA results of the samples and from published papers or public database, with predicted MS2 and iRT. With the in-silico library, the authors showed that the phosphoproteome coverage was expanded, and its promising potential in kinase signaling studies. Despite the appealing concept, there are some inherent defects, i.e. the control of FLR and FDR assessment when using unmatched libraries, which restrict the publication of the work.

RESPONSE: We deeply thank the reviewer for the concise summary of the major discovery of our work which is to expand the phosphoproteome coverage by using an *in silico* library generated with a new deep learning model. To make this conclusion more convincing, we completely agree with the reviewer that the FLR and FDR of phosphopeptide identification using our approach need to be assessed in a more strict and comprehensive manner. In addition to the target-decoy strategy originally described in the manuscript, we provided new data for the FLR/FDR assessment using three different methods in revision. The results are presented in Supplementary Figs. 5e, 7e, 9, 10c, 10d and described in the revised manuscript (Pages 11, 13, 17 highlighted in red).

Specifically, three additional methods used for the FLR/FDR assessment are:

- 1) Test predicted libraries built on a set of synthetic phosphopeptides with known site localization.
- 2) Test two-species predicted libraries comprised of human phosphopeptide precursors (positive set) and *Arabidopsis thaliana* phosphopeptide precursors (negative set).
- 3) Evaluate the false rate of phosphopeptide quantification with predicted libraries as an estimate of the FDR of phosphopeptide identification.

Detailed results are provided in our responses to the first and second comments. We hope these cross-validation tests would demonstrate the sufficient FLR/FDR control in our study, and we greatly appreciated all excellent comments from the reviewer to improve our work.

- The current version of the manuscript does not address the most significant issue in phosphorylation – localization of phosphosites in peptides. In the current work, the authors simply chose 0.75 localization confidence, referring to the work by Bekker-Jensen et al. Nevertheless, in the referred work, the library was built by directDIA, wherein the localization of phosphosites is controlled during the directDIA analysis in a way similar to typical DDA data analysis. Then the issue of FLR control in the DIA data analysis is simplified. In the current work, by introducing the hPhosSiteDB and the hPhosPepDB, there is high risk of introducing many phosphosites that do not exist in the analyzed sample. And in the case, the 0.75 localization confidence cannot guarantee a 1% FLR as expected. We have actually made some tests, and it was found that the identified phosphosites by DIA analysis highly depended on the information of phosphosites given in the spectral library. When there was a large portion of wrong phosphosites in the spectral library, the FLR from the DIA analysis results increased significantly. Without address the issue of FLR control when using non-sample specific spectral library, the downstream analysis on kinase signaling studies could include many false information. The authors need to assess the FLR using the predicted spectral libraries in comparison to DDA-based libraries.

RESPONSE: Again we thank the reviewer for making this very critical comment. First we would like to point out that in the work by Bekker-Jensen *et al.*, the authors applied a phosphosite localization confidence cut-off of 0.75 not only for data search with directDIA libraries but also for that with a project-specific DDA library (containing >70,000 phosphopeptides) and a community-based library (containing >75,000 phosphopeptides). Since our predicted libraries built on hPhosSiteDB and hPhosPepDB are of larger size than these experimental libraries, we fully agree with the reviewer that it is necessary to validate the FLR control with an independent approach beyond the target-decoy strategy.

We adopted a classical approach to FLR estimate with a synthetic phosphopeptide dataset which was also utilized in the work by Bekker-Jensen *et al.* Here we used a DIA data set acquired on 166 synthetic human phosphopeptides containing 176 clearly defined phosphosites (from *Nat Commun.* 2021 12(1):2539. PMID: 33953186). We first generated an experimental DDA library comprised of the synthetic phosphopeptide data alone (SynLib) or appended with an extensive human phosphoproteome library (RPE1 DDA). We also generated a predicted library based on the synthetic phosphopeptide information (predSynLib) and built hybrid libraries by appending the predicted synthetic phosphopeptide library with a large predicted library based on hPhosSiteDB or hPhosPepDB. Initial and iterative data searches were performed for all hybrid libraries combining the synthetic phosphopeptide library and a much larger experimental or predicted library.

As shown in Supplementary Fig. 9a, DIA data analysis with the hybrid experimental DDA library (SynLib+RPE1 DDA) gave rise to FLR <5% in both initial and iterative

searches. While data analysis with the hybrid predicted libraries (predSynLib+hPhosPepDB, predSynLib+hPhosSiteDB) yielded FLR of 6-7% in initial searches, iterative searches with the same libraries reduced FLR to 2.9% and 3.5%. In addition, iterative searches with two hybrid predicted libraries reached the maximal rate of true phosphosite recovery (94-95% TRR) that could be achieved with the pure synthetic peptide library. We also calculated the phosphosite FLR and TRR at varying site localization confidence, and found increasing the confidence cut-off did not much affect FLR for initial searches with predicted libraries unless it was set very high (>0.98) to significantly reduce the FLR yet also sacrificing the TRR (Supplementary Fig. 9b).

Therefore, this synthetic phosphopeptide data analysis implied that the FLR control for the iterative search with our predicted libraries built on public databases using the original >0.75 localization confidence is below 5% and comparable to the FLR control with the experimental DDA library. This conclusion actually agrees with the finding mentioned by the reviewer that a higher fraction of true phosphopeptides in the library would lower the FLR for DIA data analysis, which is the case for our iterative search with a focused library. Because our downstream analysis of EGF signaling pathways and regulated kinases was performed based on phosphopeptides identified from the iterative search with the hPhosPepDB predicted library, the biological finding was valid under a sufficient FLR control (FLR <3% with this assessment). We added relevant discussion and detailed methods to the revised manuscript (Pages 13, 41-42).

a

	True phosphopeptides	True phosphosites	TRR (%)	False phosphosites	FLR (%)
SynLib	154	164	93.18	0	0
predSynLib	154	164	93.18	0	0
SynLib+RPE1 DDA (initial)	136	141	80.11	5	3.42
SynLib+RPE1 DDA (iterative)	154	164	93.18	3	1.8
predSynLib+hPhosPepDB (initial)	130	133	75.57	10	6.99
predSynLib+hPhosPepDB (iterative)	156	166	94.32	5	2.92
predSynLib+hPhosSiteDB (initial)	121	124	70.45	9	6.77
predSynLib+hPhosSiteDB (iterative)	156	166	94.32	6	3.49

TRR (True recovery rate) = $N(\text{true phosphosites}) / 176$ (number of total known phosphosites)

FLR (False localization rate) = $N(\text{false phosphosites}) / (N(\text{true phosphosites}) + N(\text{false phosphosites}))$

b

Supplementary Figure 9 FLR estimation using a synthetic phosphopeptide DIA data set.

(a) Summary of true and false phosphosites identified with each library and the calculated TRR and FLR. SynLib, an experimental DDA library comprised of 166 synthetic phosphopeptides containing 176 known phosphosites; predSynLib, a predicted library built on the synthetic phosphopeptide information in SynLib; SynLib+RPE1 DDA, a hybrid experimental library combining SynLib with an extensive human phosphoproteome library RPE1 DDA; predSynLib+hPhosPepDB and predSynLib+hPhosSiteDB, hybrid predicted libraries combining predSynLib and a large predicted library built on a public database. Results are shown for the initial search with SynLib or predSynLib and initial/iterative searches with a hybrid library, all at a phosphosite localization confidence >0.75. **(b)** TRR and FLR as a function of the phosphosite localization confidence cut-off for DIA data analysis with each library listed in (a).

- The authors used “decoy” libraries with reversed peptides for library-specific FDR assessment. The reversed “decoy” library was appended to the original “target” library. The combined library was then imported into Spectronaut and used as the actual target library in the target-decoy approach performed by Spectronaut. Since the reversed peptides are too similar to the “mutated” decoys generated by Spectronaut, they are more inclined to be filtered by Spectronaut than false peptides in real samples, and therefore the FDR is underestimated. Since the predicted libraries of the hPhosSiteDB and the hPhosPepDB can include many peptides do not exist in the sample, the target-decoy approach by Spectronaut may not well control the FDR of the DIA analysis results. The authors need to evaluate FDR independently of Spectronaut’s decoy method. Also, the reviewer suggests referring to the original and manually added “decoy” libraries as other terms instead of target and decoy to avoid confusion.

RESPONSE: Spectronaut automatically assesses the FDR by creating a mutation-based decoy library and combining it with the target library so as to report identification results at a FDR <1% at both peptide and protein levels. In the original work, we manually appended a test library with a corresponding reverse-sequence library of the same size so as to evaluate library-specific FDRs which turned out to be <1% for the experimental DDA library and larger predicted libraries (Fig. 3d, Supplementary Fig. 5d). To avoid the problem of similarity between the reverse-sequence library and the Spectronaut’s mutation decoy library, we devised two software-independent metrics for assessing the phosphopeptide FDR.

First, we created two-species libraries by combining the predicted hPhosPepDB or hPhosSiteDB library and a predicted *A. thaliana* phosphoproteome library. This predicted *A. thaliana* library was generated by predictions for 103,620 precursors corresponding to 34,540 unique phosphopeptide sequences registered in PhosPhAt database (Nucleic Acids Res. 2008 D1015-21. PMID: 17984086) which compiles various DDA-based phosphoproteomics data. Figure R1 summarizes the number of phosphopeptides identified from the hPhosPepDB/hPhosSiteDB sub-library (positive set)

or from the *A. thaliana* sub-library (negative set) in the combined library from the analysis of U2OS DIA data or RPE1 DIA data. FDRs were estimated to be <1% for both DIA data sets analyzed with two predicted libraries except for the iterative search of RPE1 data with the predDIA+hPhosSiteDB library (FDR at 2.5%). These results were added to the revised manuscript as Supplementary Figs. 5e and 7e. We did not include the estimated FDR for the predDIA+hPhosSiteDB library in Supplementary Fig. 7e as this library was abandoned in the RPE1 data analysis, though the result is shown here for review only.

Figure R1 FDR estimate with a two-species library for the predDIA+hPhosPepDB library (a) or the predDIA+hPhosSiteDB library (b). Number of phosphopeptide IDs from two sub-libraries in the initial or iterative search of U2OS data or RPE1 data are indicated. FDR is calculated based on the following equation:

$$\text{FDR} = 2 \times \frac{\text{Hits}_{A. thaliana}}{\text{Hits}_{human} + \text{Hits}_{A. thaliana}}$$

Second, we evaluated the false rate of phosphopeptide quantification with experimental vs predicted libraries in the two-proteome model. If our predicted libraries could yield more falsely identified phosphopeptides, we would expect an increased percentage of phosphopeptides with quantified ratios largely deviated from the theoretical values. Supplementary Fig. 10c plots the false quantification rate (FQR) calculated under varying thresholds of the quantification error between measured and theoretical ratios of phosphopeptides identified at all dilution conditions with different libraries. These FQR curves closely overlap between the experimental DDA library (Lib 1) and any predicted library (Lib 2 to Lib 8 as defined in Fig. 6a). We also compared specific FQRs for yeast and human phosphopeptides at two quantification error thresholds (<50% or <30%) and found they were in general similar among all test libraries (Supplementary Fig. 10d). Although the absolute value of FQR cannot represent that of FDR, its relative comparison under different conditions highly suggests that FDRs for DIA data analysis with our predicted libraries are not inflated compared to that with the benchmark DDA library.

d

Yeast	Lib 1	Lib 2	Lib 3	Lib 4	Lib 6	Lib 7	Lib 8
50% quant error threshold	6.2	6.2	7.6	8.5	7.9	8.2	6.1
30% quant error threshold	16.6	16.4	18.2	19.4	19.2	19.3	16.5
Human	Lib 1	Lib 2	Lib 3	Lib 4	Lib 6	Lib 7	Lib 8
50% quant error threshold	3.7	4.2	3.3	3.5	3.1	3.2	5.3
30% quant error threshold	11.4	12.4	10.7	10.6	10.2	10.5	14.3

Supplementary Fig. 10 (c) FQR as a function of the quantification error threshold for yeast phosphopeptides (left) and human phosphopeptides (right) identified with different libraries. **(d)** FQR percentages at a 50% or 30% quantification error threshold for yeast phosphopeptides (upper) and human phosphopeptides (lower) identified with different libraries.

Taken together, these two additional FDR assessments together with the previous FLR estimate cross-validated the error rate control in our study, which is discussed in the revised manuscript (Pages 11, 13, 17). Detailed methods are also described (Pages 40, 41). As kindly suggested by the reviewer, we changed terms of the target and decoy library to the original and reverse library in revision (Pages 11, 13).

- In model training, the pre-train fine-tune pipeline is a bit weird. This pipeline is for multi-task training, while in this case, tasks are the same but on different datasets. Generally, it will be more reasonable if model is pre-trained on the combination of previous pre-trained datasets and target datasets and then fine-tuned on the target datasets.

RESPONSE: The DeepPhospho model was pre-trained with four large-scale phosphoproteomic datasets. To validate its prediction performance, we then evaluated DeepPhospho using three new datasets (two DIA and one DDA data). Each dataset was split at a ratio of 8:1:1 for model training (*i.e.* fine-tuning), validation and test separately. After the model validation, we generated DeepPhospho predicted libraries for DIA phosphoproteomics in three case studies. In each study, we first trained (*i.e.* fine-tuned) the initial pre-trained model with a specific DIA dataset and used the trained model to predict phosphopeptide fragment ion intensity and iRT in order to construct

different *in silico* libraries. Therefore, the pre-trained model serves as a foundation that will be further trained (or fine-tuned) with different target datasets. In our manuscript, training refers to “fine-tuning” the pre-trained model. We clarified this point in Methods in the revised manuscript (Page 36).

- Comparison in Figure 1b is not fair since the DeepPhospho is trained with the data from the same dataset as the test, while the others are not.

RESPONSE: In Figure 1b, we compared the performance of DeepPhospho with three other models. As pDeep2 also allows for transfer learning with a training set, we actually trained pDeep2 with the same dataset in the same way as DeepPhospho. For DeepMS2, we directly used the pre-trained model and followed the budding strategy, which was the best way for predictions as recommended by the author. For MS2PIP, we used the MS2PIP online server for predictions as recommended by the author. We described training and usage of these models in Methods (Page 37, see *Comparison of DeepPhospho with other models*)

- It is not clear how is the performance of MS2 and iRT prediction for single and multi phosphopeptides, and for S, T, Y, separately?

RESPONSE: We thank the reviewer for making this excellent comment. The prediction performance evaluated with three datasets for mono- or multi-site phosphopeptides and for phosphopeptides merely containing pS, pT or pY is summarized in Supplementary Figs. 2c, 2d. There were no significant differences in fragment ion intensity or iRT predictions between different categories of phosphopeptides. This finding was mentioned in the revised manuscript (Page 7).

Supplementary Fig 2 Evaluation of DeepPhospho with other datasets and for different categories of phosphopeptides. **(c, d)** Evaluation of DeepPhospho predictions of fragment ion intensity (c) and iRT (d) for single or multi-phosphosite peptides and for phosphopeptides merely containing pS, pT or pY. Model performance was evaluated with RPE1 DDA, RPE1 DIA and U2OS DIA data.

- The hPhosPepDB library was generated based on a phosphopeptide precursor list collected from a mass spectrometry dataset, which contains sequences, PTM sites and charge states. The hPhosSiteDB library was generated based on protein phosphosite list, which only contains sequences and PTM sites, and the library contains precursors with all the defined charge states for each phosphopeptide. The authors should give this information in the main text.

RESPONSE: To optimize the composition of the hPhosPepDB library, we initially generated 21 predicted libraries depending on the combination of precursor and fragment mass ranges, peptide length, max phosphosite number and charge state in different values. The best combination with the highest phosphoproteome coverage to generate Lib 4 and Lib 7 turned out to have charge states of 2/3/4 instead of the most

frequently observed charge state in hPhosPepDB (Supplementary Fig. 4 and Fig. 6). The hPhosSiteDB library (Lib 5) also consisted of phosphopeptides with charge states of 2/3/4. Therefore, “all three predicted libraries consist of peptide precursors with charges states of 2/3/4 whereas their sequences and phosphosites are defined by the databases”, which was added to the revised manuscript (Page 10).

- Before performing analysis using dDIA+predDDA (in Figure 3), the authors need to report the results by dDIA only.

RESPONSE: Figure R2 summarizes the identification result using the direct DIA library for the analysis of U2OS DIA data and RPE1 DIA data and comparison with the project-specific DDA library. The phosphoproteome coverage using the dDIA library was comparable to the DDA library for U2OS data search (when the DDA library was relatively small) and was lower than the DDA library for RPE1 data search (when the DDA library was quite extensive), which is a conceivable result. Because our major goal is to create different hybrid predicted libraries containing the dDIA sub-library and compare them with the benchmark DDA library, we did not include this pure dDIA library in the main library design and would greatly appreciate the reviewer’s understanding.

Figure R2 Number of phosphopeptides and phosphosites identified using a DDA or dDIA library. Percentage of the total phosphopeptide or phosphosite number is shown for the dDIA library relative to the DDA library. The proportions of shared IDs, gained IDs, lost IDs and gap IDs yielded by the dDIA library compared to the DDA library are indicated in different color. Gap IDs are those present in the DDA library yet absent in the dDIA library, thus they cannot be identified with the latter.

- Community resources of PTM sites are usually more easily accessible and have more complete coverage than mass spectrometry datasets from specific publications. Therefore, besides Lib 7 predDIA+hPhosPepDB (in Figure 3 and 4), the reviewer would like to see the performance of analysis using predDIA+hPhosSiteDB. Also, can iterative

search help to resolve the challenge of extensive query space when using libraries generated from protein phosphosite list (Lib 5 and predDIA+hPhosSiteDB)?

RESPONSE: We thank the reviewer for bringing up this interesting point. Figure R3 summarizes the result of U2OS and RPE1 data analysis with the DIA+hPhosSiteDB (Lib 5) or the predDIA+hPhosSiteDB library in parallel with other libraries in the initial and iterative search. As expected, both Lib 5 and the predDIA+hPhosSiteDB library performed much worse than the predDIA+hPhosPepDB library (Lib 7) in all cases, although the iterative search did help to increase the coverage for these two libraries (Figure R3). Because we mentioned the poor performance of hPhosSiteDB in the U2OS data analysis and abandoned it in the RPE1 data analysis, we would like to provide this result for review only.

Figure R3 Number of phosphopeptides and phosphosites identified using each library. Percentage of the total phosphopeptide or phosphosite number is shown for each predicted library relative to the project-specific DDA library (Lib 1). The proportions of shared IDs, gained IDs, lost IDs and gap IDs yielded by other libraries compared to Lib 1 are indicated in different

color. Gap IDs are those present in Lib 1 yet absent in the predicted libraries, thus they cannot be identified with the latter.

- The iterative search expands the fraction of the quantifiable phosphoproteome within the entire profiled proteome. The results suggest that the same peptide can be assigned to different features in the focus search and the general search. Then there is question on which assignment is correct.

RESPONSE: We thank the reviewer for making this critical comment. If the same peptide was assigned to different features in the initial (general) search and the iterative (focus) search, we would expect an evident discordance of its RT between the two searches. As shown in the RT correlation plot for all peptides identified in both searches from each DIA run in the RPE1 dataset, RT measurement was perfectly correlated between two searches ($R^2 > 0.999$ for all runs). This result strongly indicated the correct assignment of the same peptides to the same features in the initial and iterative searches. We described this result in the main text (Page 13, Supplementary Fig. 8).

Supplementary Figure 8 RT correlation of co-identified peptides in the initial and iterative searches of RPE1 DIA data. RT correlation is shown for peptides identified with Lib 1 **(a)** or Lib 7 **(b)** in each DIA run of the dataset.

- It shows that data mining with Lib 6 and Lib 7 uncovered 128 and 122 novel EGF-regulated phosphosites that were not revealed with Lib 1. Can these newly identified phosphosites be verified, especially with the concern of the lack of FLR control in the current method?

RESPONSE: We thank the reviewer for his/her great comment, and hope our previous FDR and FLR assessments have convinced the reviewer about the error rate control using our approach. To further support the credibility of novel regulated phosphosites uncovered in our study, we collected results from four high-quality EFG signaling proteomics projects. Critical information about the four published studies is listed as follows:

1. EasyPhos

- Publication: High-throughput and high-sensitivity phosphoproteomics with the EasyPhos platform (Nat Protoc. 2018, 13(9):1897-1916.)
- Result used: MaxQuant search result retrieved from PRIDE (EGF Stimulation_QE HFX_Figure 4_search.rar)
- Sample and data acquisition: U-87 cell line (control and EGF stimulation for 15 min); label-free DDA
- Reported regulated phosphosites: 2861

2. EGF_06

- Publication: Global, In Vivo, and Site-Specific Phosphorylation Dynamics in Signaling Networks (Cell. 2006, 127(3):635-48.)
- Result used: Supplementary Data 2
- Sample and data acquisition: SILAC-labeled HeLa cell line (EGF stimulation for 0 min, 1 min, 5 min, 10 min and 20 min); DDA
- Reported regulated phosphosites: 2487

3. CR14_EGF

- Title: Ultradeep Human Phosphoproteome Reveals a Distinct Regulatory Nature of Tyr and Ser/Thr-Based Signaling (Cell Rep. 2014, 8(5):1583-94.)
- Result used: Supplementary Table 2
- Sample and data acquisition: HeLa cell line (control and EGF stimulation for 15 min); label-free DDA
- Reported regulated phosphosites: 3959

4. LungCancerEGF_14

- Title: Identifying novel targets of oncogenic EGF receptor signaling in lung cancer through global phosphoproteomics (Proteomics. 2015, 15(2-3):340-55.)
- Result used: Supplementary Table 1
- Sample and data acquisition: SILAC-labeled H3255/H1975 cell lines (control and EGF stimulation for 3 min); DDA
- Reported regulated phosphosites: 2060

We then summarize the numbers of total regulated phosphosites (blue) and those also reported in the previous studies (red series) revealed with the experimental DDA library (Lib 1) or the two predicted libraries (Lib 6 and Lib 7) in Figure R4a. Numbers of the novel regulated phosphosites only revealed by Lib 6 or Lib 7 and overlapping with those reported in previous studies are also shown below (two groups on the right, Figure R4a). The cumulative novel EGF-regulated phosphosites that were repeatedly found in previous studies are 63 and 67, nearly or above half of the total novel phosphosites revealed by Lib 6 and Lib 7 (Figure R4b). Moreover, data mining with the two predicted libraries uncovered more regulated phosphosites than Lib 1 (331/317 vs 271) with a percentage of verifiable sites very similar to Lib 1 (Figure R4b). These results highly support the credibility of regulated phosphosites uncovered with the predicted libraries.

Figure R4 (a) Number of total regulated phosphosites and those also reported in each previous study revealed with the DDA library (Lib 1) or two predicted libraries (Lib 6 and Lib 7). Novel regulated phosphosites revealed by Lib 6 or Lib 7 and reported in the

previous study are also shown. (b) The cumulative number of regulated phosphosites reported in previous studies (red) and number of total regulated phosphosites revealed with each library (blue).

Minor:

- The PCC scores between the spectra in Figure 2c should be indicated.

RESPONSE: PCC scores are indicated in Fig. 2c and Supplementary Fig. 3b.

- It should be stated in the title that deep learning is used for spectral library generation.

RESPONSE: The title has been changed to “DeepPhospho: accelerate DIA phosphoproteome profiling through *in silico* library generation”.

- The Spectronaut spectral libraries (.kit) and search results (.sne) files should be provided in the ProteomeXchange repository.

RESPONSE: These files are provided in the ProteomeXchange repository (Identifier PXD025112).

Reviewer #2 (Remarks to the Author):

Summary

The authors describe a novel neural network (nn) architecture that allows to predict fragment ion intensities and indexed retention time (iRT) for phosphorylated peptides. The authors show that predictions with this nn outperform published nn. The authors suggest a novel workflow where the nn network is trained (transfer learning) for a setup using a library derived from searching the DIA data directly. The peptides from a database and from the spectrum-centric DIA search are used to defined the precursors to predict. The authors show that with this workflow they can outperform workflows based on DDA libraries while saving extra measurement time that would otherwise be spent on the DDA runs. The research presented is relevant and of sufficient quality for NC. The manuscript is relatively dense and sometimes difficult to follow. My main concerns are listed below under major points. The authors show metrics that were achieved by applying transfer learning to the model presented. However, using the web interface such transfer learning cannot be applied to the model. Hence, the performance that users of the web interface may experience can differ from

the one presented in this manuscript. Minimally the authors should show the difference of performance in the respective figures (see below). Ideally, the web interface would also allow transfer learning by uploading a library.

RESPONSE: We very much thank the reviewer for his/her positive comments on our work and the great suggestion of adding the transfer learning function to our DeepPhospho web interface. Unfortunately, building this online function will consume a tremendous computational resource which is not affordable for our lab. Alternatively, we have created an offline DeepPhospho app for users who are interested in doing transfer learning with their own datasets. This offline app is a full wrapper of our DeepPhospho pipeline and can be easily launched on a desktop. It allows users to directly use the pre-trained model, train a new model, or fine-tune the model parameters with their own target datasets before making predictions with a selected model. Using our offline DeepPhospho app, a ready-to-use spectral library will be generated as an output file. The offline app now supports model training using result files in the Spectronaut library format or in the MaxQuant msms format, and it supports fragment ion intensity and iRT prediction using input files in four different formats (Spectronaut style, MaxQuant style, Comet style, and DeepPhospho self-defined style as specified in our DeepPhospho web interface). The offline DeepPhospho app can be downloaded from DeepPhospho GitHub repository at <https://github.com/weizhenFrank/DeepPhospho> and its release is announced in our DeepPhospho web interface.

We added the details of the offline app to Methods (Page 44).

Major Points

- Fig 1 Please add all metrics also for the purely pre-trained model without the transfer learning. Users of the web interface will not be able to perform the transfer learning and hence will see a performance that is more similar to the pre-trained model.

RESPONSE: We thank the reviewer for making this critical comment. Evaluation of the pre-trained models with three separate datasets has been added to the revised manuscript (Supplementary Figs. 11a and 11b, also shown below). Conceivably, predictions with the pre-trained models are not as accurate as the trained models.

Supplementary Fig. 11 (a) Evaluation of the pre-trained fragment ion intensity model based on PCC (left) and SA (right) analysis with three test sets. **(b)** Evaluation of the pre-trained iRT model based on iRT correlation analysis with three test sets. To deal with chromatography variation in different data sets, we randomly selected ten peptide-iRT pairs at five iRT percentiles (10%, 30%, 50%, 70%, 90%) and calibrated the predicted iRTs by second-order polynomial fitting.

- page 7, line 161: There are other potential reasons why the empirical and predicted spectra agree better for DDA. For instance collision energy is determined more homogeneously for precursors in DDA as compared to DIA. In DIA collision energies cannot be chosen optimally for all precursors in a precursor selection window. FDR control in spectrum-centric searching is very established and hence FDR can be assumed to be similar between DDA and DIA libraries. However, it is a fair point that ~1% of entries in the libraries are wrong and could be corrected by prediction. I'd suggest to re shape the text in a way that does not imply that FDR in DIA libraries is generally elevated but rather point to the fact that prediction has the advantage that there are no more "wrong" assays. Currently, there is not enough evidence in the manuscript that would suffice to indicate to an elevated FDR (as mentioned above lower SA or PCC can have other reasons). I'd suggest to move everything related to that point into the supplementary.

RESPONSE: We fully agree with the reviewer that no evidence has suggested a higher FDR using the direct DIA library than the conventional DDA library. In fact, we and others have performed various FDR assessments to prove the experimental DDA/DIA libraries and the predicted libraries all have a sufficient FDR control (please see our response to your last major point about FDR estimation). We apologize for the misleading description in the original manuscript and have toned down the text to only point out the usage of spectral prediction to pinpoint possibly "wrong" assignments in an experiment library (Pages 7, 8).

- The provided webpage does not allow transfer learning. Ideally the web service would also allow transfer learning by uploading a spectral library.

RESPONSE: Please see our response to the reviewer's summary. We provide an offline app to allow transfer learning based on DIA or DDA search results.

- page 39, line 887: In the spectrum-centric analysis the basic assumption that false identifications will uniformly distribute over target and decoy database provide a simple framework for FDR estimation (Elias and Gygi et al. 2007). This simple basic assumption is not transferrable to the peptide-centric analysis. Please state in the methods text that this formula can only serve as a rough estimate in the case of peptide-centric analysis.

RESPONSE: We thank the reviewer for making this excellent comment. We acknowledged the rough estimate of FDR for DIA data analysis with the target-decoy strategy in the revised manuscript (Page 41, first paragraph). In addition, to verify the FDR and FLR (false localization rate) control of phosphopeptide identification in our study, we adopted three additional methods in revision as follows:

- 1) Test predicted libraries built on a set of synthetic phosphopeptides with known site localization.
- 2) Test two-species predicted libraries comprised of human phosphopeptide precursors (positive set) and *Arabidopsis thaliana* phosphopeptide precursors (negative set).
- 3) Evaluate the false rate of phosphopeptide quantification with predicted libraries as an estimate of the FDR of phosphopeptide identification.

Results for the FLR/FDR assessment using three different methods are presented in Supplementary Figs. 5e, 7e, 9, 10c, 10d and described in the revised manuscript (Pages 11, 13, 17). Please also refer to our point-to-point responses to the first and second comments raised by Reviewer 1 for more details.

Minor Points

- page 4, line 84: Please better differentiate point 1) from 2)

RESPONSE: We thank the reviewer for revealing the redundancy of points 1) and 2), and rephrased them to be one central question: "Can DeepPhospho predicted libraries outperform the benchmark experimental DDA library to enable faster and deeper DIA phosphoproteome profiling?" (Page 4).

- page 5, line 93: Please mention the number of parameters of the main model used in the main text and the number of parameters of all models tested in the supplementary. Ideally compare to number of parameters of other models presented in literature and benchmarked in this study.

RESPONSE: We summarize the number of these parameters for DeepPhospho and other models in Supplementary Fig. 11c (shown below).

c

	Number of parameters (M)
DeepPhospho (LSTM+Transformer)	37.84
pDeep2	7.04
DeepMS2	0.21
DeepPhospho RT Ensemble	168.16
LSTM (ablation study)	21.02
Transformer (ablation study)	6.33
CNNTransformer (ablation study)	46.29

Supplementary Fig. 11 (c) Total number of model parameters in DeepPhospho, pDeep2, DeepMS2 and three models assessed in the ablation study.

- page 6, line 134: Please mention the total number of precursors used for the pre-training.

RESPONSE: It was specified in Supplementary Fig. 11d (shown below).

d

Number of precursors used for ion intensity model pre-training		
	Precursors for training	Precursors for test
Mouse brain DDA data	80,494	8,945
VeroE6 DIA data	48,602	5,402
Yeast DIA data	38,980	4,332

Number of peptides used for retention time model pre-training		
	Peptides for training	Peptides for test
Human phosphopeptide RT data	184,102	20,456
Mouse brain DDA data	64,172	3,378
VeroE6 DIA data	39,064	4,341
Yeast DIA data	32,227	3,581

Supplementary Fig. 11 (d) Number of precursors and peptides used for DeepPhospho pre-training.

- Please better explain in the methods how peptides with n phosphosites were spanned (col 6, suppl fig 4)

RESPONSE: We apologize for missing this information in the original manuscript. To optimize the predicted hPhosPepDB library, we built different libraries containing peptides with different site numbers as registered in the database. The max site number (1, 2 or 3) in Supplementary Figs. 4, 6 indicates the max number of phosphosites present in all peptides in the library. A max site number of 1 indicates only mono-site phosphopeptides are included in the library while a max site number of 3 indicates peptides with 1-3 phosphosites are all included. For the predicted hPhosSiteDB library, we set the max phosphosite number in each peptide to 1, meaning that only mono-site phosphopeptides inferred from the database were included in the library while multi-site phosphopeptides were not considered. The max site number for each library was clarified in the legend of Supplementary Fig. 4.

- page 12, line 280: Please also indicate the library type and size in a sub figure similar to fig 3

RESPONSE: Figure 4a was revised to describe the main library design (*i.e.* library type and composition) for RPE1 DIA data analysis. The library size of each initial library and the corresponding focused library is indicated in Supplementary Fig. 7b.

- page 12, line 283: Please better explain the iterative search and the metrics used. On how many runs was the initial search performed? Were average figures used for the number of phosphopeptides and sites reported (otherwise I wouldn't understand how the numbers can go up in the iterative search strategy)?

RESPONSE: In the RPE1 DIA data analysis, either the initial search or the iterative search was performed only once for 18 runs of DIA data combined together. Thus there were no average numbers from "multi-run searches". The iterative search used a focused library that only contained peptide precursors identified from the initial search with a specific experimental or predicted library. We speculated that the higher fraction of "true phosphopeptides" existing in the analyzed samples in the focused library would increase the numbers of quantifiable phosphopeptides/sites from the iterative search.

- please mention training hardware (GPUs?) and time (hours, epoches) in methods.

RESPONSE: The models were trained either on 4 TITAN Xp GPUs or 4 GTX1080Ti GPUs. The training time is related to the size of a dataset. For the fragment ion intensity model, it took about 12 hours for pre-training with a dataset containing 80k precursors, and about 2 hours for fine-tuning with a dataset containing 40k precursors. For the iRT model, it took about 22 hours for pre-training with a dataset containing 180k peptides, and about 4 hours for fine-tuning with a dataset containing 30k peptides.

This information was added to Methods in revision (Page 36).

REVIEWER COMMENTS

Reviewer #1 (Remarks to the Author):

The authors have revised their manuscript. The major changes include FLR estimation by using a dataset of synthetic phosphopeptides and FDR assessment using a two-species strategy. The authors also use the FQR to support the accuracy in identification. Nevertheless, there are still concerns on the FLR and FDR assessments. Without experimental knowledge, there is always a high risk of large FLR and FDR. Therefore, the reviewer would like to see more demonstration on the performance of FLR and FDR control.

- For the FLR assessment using the dataset of synthetic phosphopeptides, please provide more information on the SynLib+RPE1 DDA, predSynLib+hPhosPepDB, and predSynLib+hPhosSiteDB libraries. The SynLib should contain only the phosphopeptides with correct phosphosite. Then, the question is how many false phosphosites are included in the three hybrid libraries. The information is important to assess the FLR control performance in real case. The authors should also include more datasets of synthetic phosphopeptides that are public available from previous publications to further demonstrate the performance of the FLR control.

- For FDR assessment using the two-species strategy, the percentage of $2 * \text{Hits}(A. \text{ thaliana}) / (\text{Hits}(A. \text{ thaliana}) + \text{Hits}(\text{human}))$ is not the FDR. The hPhosPepDB contains 204,606 human phosphopeptides; the hPhosSiteDB contains 350,719 phosphopeptide; while the A. thaliana library contains only 34,540 phosphopeptides. As the library size of entrapment species is only $\sim 1/7$ or $1/10$ of the target species, the FDR is largely underestimated. Actually, the FDR estimation by using entrapment species is complicated. When using an entrapment strategy, the authors have to consider that a false hit is equally likely to come from the entrapment assay library and the part of the target assay library that is not present in the sample (to be estimated by target assay library multiplied by π_0 , Nat. Methods 14, 921-927 (2017)). Therefore, the entrapment peptides only constitute a part of the false positives.

- On Figure R1, the iterative search leads to a higher "estimated FDR", which is surprising. How is the size of the library during the iterative search for both human phosphopeptides and the A. thaliana phosphopeptides. I suppose that in the library for iterative search the number of human phosphopeptides is closer to the number of A. thaliana phosphopeptides compared to the initial search, thereby the "estimated FDR" in the iterative search is higher and closer to the real FDR. It should be noted that in the iterative search the target library for human phosphopeptides has already been optimized. Therefore, a higher FDR in the initial search is always expected compared to

the iterative search. It is noticed that the percentage of entrapment in iterative search can reach 3.2%. A higher real FDR in the initial search is then expected.

In addition to the questions on FDR and FLR control, there are also some other concerns to be solved.

- The performance of DeepPhospho is better than pDeep2, DeepMS2 and MS2PIP. It can be resulted from the effect that the deep learning framework itself is better or the transfer learning is applied. To clarify this effect, the reviewer would like to see the performance of DeepPhospho without transfer learning.

- Supplementary Figure 2, mono-phosphosite and only pS gives much better performance than the others. Please comment on this performance. Is it due to the fact that there are more data available during model training for pS and mono-phosphosite.

- The results by dDIA and dDIA+DDA should be added for benchmark in Figure 3, 4 and 6. In the current version, the authors state that predDIA+predDDA or predDIA+hPhosPepDB provides the best performance. When considering DDA based experimental library workflow, it is necessary to compare the performance of those predicted library to that of dDIA+DDA.

- The bar plots in Figure 3c, 4c and 4d are misleading. It looks like all the other libraries would lead to more identifications than Lib1. This is not true for Lib5 (Gain < Loss + Gap). The reviewer suggests to plot the Loss and Gap in the negative direction, and move the Gain adjacent to the Shared, like Figure 4b in the publication of Prosit (DOI: 10.1038/s41592-019-0426-7).

- The authors mentioned in the introduction that “we first developed a fundamentally new deep learning framework, termed DeepPhospho, to achieve highly accurate predictions for phosphopeptides”. This is misleading. The deep learning framework itself is not fundamentally new. It is modified from the ones used in the natural language processing. It is newly used to the proteome data analysis.

- On page 10, the authors mentioned “which underlies the importance of selecting an appropriate database and optimizing the library construction parameters in the performance of predicted libraries built on public databases”. More discussion is expected here on the choice of database. Is the hPhosPepDB applicable to any datasets from human tissue samples/body fluid samples and human cell lines. How shall the users optimize the database for PTM sites information.

- In figure 6c, results of Lib 6 should be added to compare the performance between purely predicted based and DDA pre-knowledge-based libraries.

Reviewer #2 (Remarks to the Author):

Thanks, the authors have addressed all my concerns.

REVIEWER COMMENTS

Reviewer #1 (Remarks to the Author):

The authors have revised their manuscript. The major changes include FLR estimation by using a dataset of synthetic phosphopeptides and FDR assessment using a two-species strategy. The authors also use the FQR to support the accuracy in identification. Nevertheless, there are still concerns on the FLR and FDR assessments. Without experimental knowledge, there is always a high risk of large FLR and FDR. Therefore, the reviewer would like to see more demonstration on the performance of FLR and FDR control.

RESPONSE: We deeply thank the reviewer for recognizing our previous revision and pointing out specific concerns to further improve our work. In this second revision, we made more efforts in demonstrating our FLR and FDR control, and would very much appreciate the reviewer's expert comments.

- For the FLR assessment using the dataset of synthetic phosphopeptides, please provide more information on the SynLib+RPE1 DDA, predSynLib+hPhosPepDB, and predSynLib+hPhosSiteDB libraries. The SynLib should contain only the phosphopeptides with correct phosphosite. Then, the question is how many false phosphosites are included in the three hybrid libraries. The information is important to assess the FLR control performance in real case. The authors should also include more datasets of synthetic phosphopeptides that are public available from previous publications to further demonstrate the performance of the FLR control.

RESPONSE: Shown below are the numbers of total false phosphopeptides and phosphosites that are present in the two predicted libraries (hPhosPepDB and hPhosSiteDB) as well as in the experimental library (RPE1 DDA). The identified false phosphosites in the iterative search by using different libraries (shown in Supplementary Fig. 9) are also listed below.

Library	False phosphopeptides	False phosphosites	Identified false phosphosites
hPhosSiteDB	186	169	6
hPhosPepDB	223	136	5
RPE1 DDA	149	61	3

When we were looking for additional synthetic phosphopeptide datasets from previous publications, unfortunately we did not find a suitable one for synthetic human phosphopeptides which has to be acquired in the DIA mode with the peptide sequence and site information all clearly defined. Instead we found a synthetic yeast

phosphopeptide dataset from the publication *Nat Biotechnol* 35, 781-788 (2017). This SWATH-MS dataset was acquired on 300 synthetic yeast phosphopeptides (containing 321 phosphosites) in the background of human peptides identified by DDA analysis of the same sample. So we first generated an experimental DDA library comprised of the synthetic phosphopeptide data alone (SynLib) or appended with an experimental yeast phosphoproteome library (Yeast DDA). We also generated a predicted library based on the synthetic phosphopeptide information (predSynLib) and built a hybrid library by appending the predSynLib library with a large predicted library yPhosPepDB. This predicted yPhosPepDB library was built on phosphopeptides reported in a deep yeast phosphoproteomic study and it was also evaluated in our two-proteome model (Figure 6).

The FLRs estimated with this dataset for different libraries are summarized below.

Library	True phosphopeptides	True phosphosites	False phosphosites	FLR (%)
SynLib	253	260	0	0
predSynLib	241	248	0	0
SynLib+Yeast DDA (initial)	153	157	4	2.48
SynLib+Yeast DDA (iterative)	257	265	2	0.75
predSynLib+yPhosPepDB (initial)	179	184	12	6.12
predSynLib+yPhosPepDB (iterative)	252	261	8	2.97

The total number of false phosphopeptides and phosphosites in the two libraries is listed in the table below.

Library	False peptides	False sites
Yeast DDA	418	237
yPhosPepDB	903	508

As the number of total false phosphosites in the yPhosPepDB library (508) is much higher than that in the experimental Yeast DDA library (237), data search using the predSynLib+yPhosPepDB library gave rise to more false sites (8 sites) than the SynLib+Yeast DDA library (2 sites), yet the FLR for the predicted library in the iterative search was still at 3.0%, which is comparable to the FLRs estimated for two predicted human phosphoproteome libraries (2.9% and 3.5%) as shown in our previous response letter. Taken together, using both the human and yeast synthetic peptide data, we have demonstrated the sufficient FLR control with our predicted library strategy.

- For FDR assessment using the two-species strategy, the percentage of $2 \times \text{Hits}(A. thaliana) / (\text{Hits}(A. thaliana) + \text{Hits}(\text{human}))$ is not the FDR. The hPhosPepDB contains 204,606 human phosphopeptides; the hPhosSiteDB contains 350,719 phosphopeptide;

while the *A. thaliana* library contains only 34,540 phosphopeptides. As the library size of entrapment species is only ~1/7 or 1/10 of the target species, the FDR is largely underestimated. Actually, the FDR estimation by using entrapment species is complicated. When using an entrapment strategy, the authors have to consider that a false hit is equally likely to come from the entrapment assay library and the part of the target assay library that is not present in the sample (to be estimated by target assay library multiplied by π_0 , *Nat. Methods* 14, 921-927 (2017)). Therefore, the entrapment peptides only constitute a part of the false positives.

RESPONSE: We very much thank the reviewer for pointing out the mistake in our FDR calculation with the two-species library. According to the work mentioned above (*Nat. Methods* 14, 921-927 (2017)), we have modified the formula for FDR assessment as below.

$$\text{NormHits}_{False} = \text{Hits}_{False} \times \frac{\text{LibSize}_{Human}}{\text{LibSize}_{A. thaliana}} \times \frac{1}{1 - 0.01}$$

$$\text{FDR} = \frac{\text{NormHits}_{False} + \text{Hits}_{True}}{\text{NormHits}_{False}}$$

where LibSize_{Human} is the number of all phosphopeptides in the human sub-library, and $\text{LibSize}_{A. thaliana}$ is the number of all phosphopeptides in the *A. thaliana* sub-library. The constant item $\frac{1}{1-0.01}$ is multiplied to offset the inherent error tolerance of 1% FDR for all results.

Figure R1 lists the new FDRs calculated for different datasets and different predicted libraries, which are below 3% for both hPhosPepDB and hPhosSiteDB libraries in the initial search and drop to <0.1% for two libraries in the iterative search.

Figure R1 FDR estimate with a two-species library for the predDIA+hPhosPepDB library (a) or the predDIA+hPhosSiteDB library (b). Number of phosphopeptide IDs from two sub-libraries in the initial or iterative search of U2OS data or RPE1 data are indicated. FDR is re-calculated using the corrected formula described above.

The significant decline of FDR in the iterative search is attributed to the focused libraries of a much smaller size than the initial libraries. This result actually agreed with

the finding from publication *Nat. Methods* 14, 921-927 (2017), which showed that a sample-matching and focused library outperformed a more comprehensive library. We have updated the FDR results in Supplementary Figures 5e and 7e, and revised Methods (Page 39, highlighted in red).

- On Figure R1, the iterative search leads to a higher “estimated FDR”, which is surprising. How is the size of the library during the iterative search for both human phosphopeptides and the *A. thaliana* phosphopeptides. I suppose that in the library for iterative search the number of human phosphopeptides is closer to the number of *A. thaliana* phosphopeptides compared to the initial search, thereby the “estimated FDR” in the iterative search is higher and closer to the real FDR. It should be noted that in the iterative search the target library for human phosphopeptides has already been optimized. Therefore, a higher FDR in the initial search is always expected compared to the iterative search. It is noticed that the percentage of entrapment in iterative search can reach 3.2%. A higher real FDR in the initial search is then expected.

RESPONSE: When we used the corrected formula described above to re-calculate FDRs, the estimated FDRs for the iterative search were indeed much lower than those for the initial search (Figure R1), which fits the reviewer’s expectation.

In addition to the questions on FDR and FLR control, there are also some other concerns to be solved.

- The performance of DeepPhospho is better than pDeep2, DeepMS2 and MS2PIP. It can be resulted from the effect that the deep learning framework itself is better or the transfer learning is applied. To clarify this effect, the reviewer would like to see the performance of DeepPhospho without transfer learning.

RESPONSE: We thank the reviewer for making this critical comment. Evaluation of the pre-trained models with three separate datasets are available in the revised manuscript (Supplementary Figs. 13a and 13b, also shown below). Conceivably, predictions with the pre-trained models are not as accurate as the trained models.

Supplementary Fig. 13 (a) Evaluation of the pre-trained fragment ion intensity model based on PCC (left) and SA (right) analysis with three test sets. **(b)** Evaluation of the pre-trained iRT model based on iRT correlation analysis with three test sets. To deal with chromatography variation in different data sets, we randomly selected ten peptide-iRT pairs at five iRT percentiles (10%, 30%, 50%, 70%, 90%) and calibrated the predicted iRTs by second-order polynomial fitting.

- Supplementary Figure 2, mono-phosphosite and only pS gives much better performance than the others. Please comment on this performance. Is it due to the fact that there are more data available during model training for pS and mono-phosphosite.

RESPONSE: We have added a new panel 2e to Supplementary Figure 2 to summarize the total number of mono- or multi-phosphosite peptides and phosphopeptides merely containing pS, pT or pY in each library. As expected by the reviewer, the larger fractions of mono-phosphosites and pS-only peptides may explain why they give better performance than the others. We also revised the text on Page 7.

e

	Number of precursors in training set					Number of peptides in training set				
	Mono-phosphosite	Multi-phosphosites	Only pS	Only pT	Only pY	Mono-phosphosite	Multi-phosphosites	Only pS	Only pT	Only pY
RPE1 DDA	70,805	23,260	72,648	11,904	2,331	53,310	18,247	54,425	9,407	1,854
RPE1 DIA	13,569	5,848	15,102	2,217	296	11,195	4,978	12,399	1,926	249
U2OS DIA	27,682	6,034	25,527	5,780	336	21,977	4,940	20,236	4,653	299

Supplementary Figure 2(e) Number of precursors used for training the fragment ion intensity model (left) and number of phosphopeptides used for training the iRT model (right). Phosphopeptides in different categories are separately analyzed.

- The results by dDIA and dDIA+DDA should be added for benchmark in Figure 3, 4 and 6. In the current version, the authors state that predDIA+predDDA or predDIA+hPhosPepDB provides the best performance. When considering DDA based experimental library workflow, it is necessary to compare the performance of those predicted library to that of dDIA+DDA.

RESPONSE: We thank the reviewer for making this great suggestion. In our initial study design, we used an experimental DDA library (Lib 1) widely applied to DIA phosphoproteomics data analysis as the benchmark to be compared with different predicted or hybrid libraries (Lib 2-8). Adding the results of two other experimental libraries (dDIA and dDIA+DDA) would dramatically change the study design, making it very complicated and out of focus when investigating 9-10 libraries altogether. And we would need to renumber all the libraries to be evaluated. To keep the original design, we would like to present the comparison of search results using DIA and DIA+DDA libraries vs using predDIA+predDDA or predDIA+hPhosPepDB libraries in a new figure (Supplementary Fig. 12, shown below).

Supplementary Figure 12 Number of phosphopeptides and phosphosites identified using two other experimental libraries in comparison to Lib 1, Lib 6 and Lib 7. DIA and DIA+DDA library refer to the direct DIA library and the merged DIA and DDA library respectively, both built on the experimentally acquired DIA or DDA MS data. The initial search result is shown for the U2OS data while the iterative search results are shown for RPE1 and two-proteome data. The proportions of shared identifications (IDs), gained IDs, lost IDs and gap IDs yielded by different libraries compared to Lib 1 are indicated in different color.

The following statement was added to the revised manuscript (Page 19): “Notably, we used an experimental DDA library as the benchmark here to be compared with different predicted libraries. When we built a larger experimental library by merging the direct DIA and DDA libraries, the advantage of predicted libraries to increase the phosphoproteome coverage remained in all datasets evaluated in this study (Supplementary Fig. 12)”.

To keep it concise, we only mentioned the DIA+DDA library here as it always outperforms the DDA library or the direct DIA library. We would deeply appreciate the reviewer’s kindest understanding of our technical difficulty in adding the results directly to main figures.

- The bar plots in Figure 3c, 4c and 4d are misleading. It looks like all the other libraries would lead to more identifications than Lib1. This is not true for Lib5 (Gain < Loss + Gap). The reviewer suggests to plot the Loss and Gap in the negative direction, and move the Gain adjacent to the Shared, like Figure 4b in the publication of Prosit (DOI: 10.1038/s41592-019-0426-7).

RESPONSE: All related main figures and supplementary figures are reformatted as kindly suggested by the reviewers.

- The authors mentioned in the introduction that “we first developed a fundamentally new deep learning framework, termed DeepPhospho, to achieve highly accurate predictions for phosphopeptides”. This is misleading. The deep learning framework itself is not fundamentally new. It is modified from the ones used in the natural language processing. It is newly used to the proteome data analysis.

RESPONSE: We agree with the reviewer and have deleted “fundamentally” in this sentence (Page 4). DeepPhospho can be regarded a new deep learning model of which the framework is developed based on Transformer and different from all published models for spectral prediction.

- On page 10, the authors mentioned “which underlies the importance of selecting an appropriate database and optimizing the library construction parameters in the performance of predicted libraries built on public databases”. More discussion is expected here on the choice of database. Is the hPhosPepDB applicable to any datasets from human tissue samples/body fluid samples and human cell lines. How shall the users optimize the database for PTM sites information.

REPOSE: For the hPhosPepDB library, we first generated 21 predicted libraries depending on the combination of precursor and fragment mass ranges, peptide length, max phosphosite number and charge state in different values. Then the best combination giving rise to the highest phosphoproteome coverage was used to prepare Lib 4 and Lib 7. Notably, the best combination of library parameters for the U2OS DIA dataset vs the RPE1 DIA dataset was different (Supplementary Figs. 4, 6). That's why we mentioned the importance of "optimizing the library construction parameters in the performance of predicted libraries built on public databases". As for the applicability of the hPhosPepDB library to any datasets from human tissue samples/body fluid samples and human cell lines, it is quite an interesting point and would need more investigations. So we added new discussion to the revised manuscript (Page 20).

- In figure 6c, results of Lib 6 should be added to compare the performance between purely predicted based and DDA pre-knowledge-based libraries.

RESPONSE: Result of Lib 6 has been added to Figure 6c (shown below), which largely outperforms the benchmark DDA library yet not as well as Lib 8 that was built on the public human phosphoproteome database hPhosPepDB.

Figure 6(c) Number of phosphopeptides and phosphosites from the human proteome that were quantified from the iterative search using each library.

REVIEWERS' COMMENTS

Reviewer #1 (Remarks to the Author):

In this revision, the authors have addressed my major concerns. I think now the manuscript provides valuable information to the society of proteome research and brings a useful tool for the study of protein phosphorylation. I have only some minor concerns about the presentation of the work in the main text as well as SI:

1. In the rebuttal letter, the authors provide information on the false phosphopeptides and false phosphosites in different libraries for human dataset and yeast dataset as tables. These should be included in the SI for publication. As I mentioned in the previous round of review, the information is very important to assess the FLR control in real case, and thus should be provided to the readers instead of reviewers only.
2. The results on the dataset of synthetic yeast phosphopeptides should be included in the SI for publication to support the FLR control performance of the method.
3. The equation for FDR calculation in the rebuttal letter is wrong but correct the main manuscript. So, if the rebuttal letter will be published, please correct this mistake.
4. Now the FDR estimation is much more reasonable. However, the ratio of $\text{NormHits_False}/(\text{NormHits_False}+\text{Hits_True})$ calculated by the authors is not equal to FDR, because in the human sub-library there are also phosphopeptides not included in the sample, i.e. the human sub-library is larger than true target library, and this part should be removed from the numerator and added to the denominator of the equation when calculating the NormHits_false, leading to a smaller value of the NormHits_false. Such modification will indeed result in a smaller value of the FDR. Since it is hard to estimate the number of phosphopeptides in the human sub-library that is not present in the sample, it is difficult to calculate an accurate FDR when using the entrapment strategy. This will not change the conclusion of the work, but I would suggest to use some other name instead of estimated FDR for this calculated ratio in the current work, or at least add a paragraph of discussion to clarify the issue.

Reviewer #1 (Remarks to the Author):

In this revision, the authors have addressed my major concerns. I think now the manuscript provides valuable information to the society of proteome research and brings a useful tool for the study of protein phosphorylation. I have only some minor concerns about the presentation of the work in the main text as well as SI:

1. In the rebuttal letter, the authors provide information on the false phosphopeptides and false phosphosites in different libraries for human dataset and yeast dataset as tables. These should be included in the SI for publication. As I mentioned in the previous round of review, the information is very important to assess the FLR control in real case, and thus should be provided to the readers instead of reviewers only.

RESPONSE: We thank the reviewer for his/her kind suggestion and have added our analysis of two synthetic phosphopeptide datasets to Supplementary Figure 9. The new result for the yeast dataset is briefly described in the revised manuscript (Page 13, highlighted in red).

2. The results on the dataset of synthetic yeast phosphopeptides should be included in the SI for publication to support the FLR control performance of the method.

RESPONSE: This result is presented in Supplementary Figure 9 (c and d).

3. The equation for FDR calculation in the rebuttal letter is wrong but correct the main manuscript. So, if the rebuttal letter will be published, please correct this mistake.

RESPONSE: We apologize for this mistake which is corrected in the submitted file "response letter_2nd rev_corrected".

4. Now the FDR estimation is much more reasonable. However, the ratio of $\text{NormHits_False}/(\text{NormHits_False}+\text{Hits_True})$ calculated by the authors is not equal to FDR, because in the human sub-library there are also phosphopeptides not included in the sample, i.e. the human sub-library is larger than true target library, and this part should be removed from the numerator and added to the denominator of the equation when calculating the NormHits_false , leading to a smaller value of the NormHits_false . Such modification will indeed result in a smaller value of the FDR. Since it is hard to estimate the number of phosphopeptides in the human sub-library that is not present in the sample, it is difficult to calculate an accurate FDR when using the entrapment strategy. This will not change the conclusion of the work, but I would suggest to use some other name instead of estimated FDR for this calculated ratio in the current work, or at least add a paragraph of discussion to clarify the issue.

RESPONSE: We thank the reviewer for making this critical point. As mentioned by the reviewer, the equation we used here possibly overestimated the true FDR yet our estimated FDRs were still below 3% for different datasets (Supplementary Figures 5e and 7e). According to the reviewer's great suggestion, we discussed the difficulty of accurate FDR calculation with the entrapment strategy in the revised manuscript as follows: "Notably, it is difficult to calculate an accurate FDR using the entrapment strategy given that it is unlikely to know the exact number of phosphopeptides in the human sub-library that is not present in the sample. This metric is

provided to roughly estimate and compare the error rates of phosphopeptide identification when using different libraries.” (Page 32).